# ANYPREFER: AN AGENTIC FRAMEWORK FOR PREFERENCE DATA SYNTHESIS

**Yiyang Zhou**[1][*]   **Zhaoyang Wang**[1*]   **Tianle Wang**[1*]   **Shangyu Xing**[1†]   **Peng Xia**[1†]   **Bo Li**[1†]
**Kaiyuan Zheng**[3†]   **Zijian Zhang**[1†]   **Zhaorun Chen**[4]   **Wenhao Zheng**[1]   **Xuchao Zhang**[5]
**Chetan Bansal**[5]   **Weitong Zhang**[1]   **Ying Wei**[2]   **Mohit Bansal**[1]   **Huaxiu Yao**[1]
[1]UNC-Chapel Hill   [2]NTU   [3]University of Washington
[4]UChicago   [5]Microsoft Research
{yiyangai,zhaoyang,huaxiu}@cs.unc.edu

## ABSTRACT

High-quality preference data is essential for aligning foundation models with human values through preference learning. However, manual annotation of such data is often time-consuming and costly. Recent methods often adopt a self-rewarding approach, where the target model generates and annotates its own preference data, but this can lead to inaccuracies since the reward model shares weights with the target model, thereby amplifying inherent biases. To address these issues, we propose `Anyprefer`, a framework designed to synthesize high-quality preference data for aligning the target model. `Anyprefer` frames the data synthesis process as a cooperative two-player Markov Game, where the target model and the judge model collaborate together. Here, a series of external tools are introduced to assist the judge model in accurately rewarding the target model's responses, mitigating biases in the rewarding process. In addition, a feedback mechanism is introduced to optimize prompts for both models, enhancing collaboration and improving data quality. The synthesized data is compiled into a new preference dataset, `Anyprefer-V1`, consisting of 58K high-quality preference pairs. Extensive experiments show that `Anyprefer` significantly improves model alignment performance across four main applications, covering 21 datasets, achieving average improvements of 18.55% in five natural language generation datasets, 3.66% in nine vision-language understanding datasets, 30.05% in three medical image analysis datasets, and 16.00% in four visuo-motor control tasks.

## 1 INTRODUCTION

Foundation models, including large language models (LLMs) and large vision-language models (LVLMs), have greatly enhanced AI model's ability to understand text, interpret images, and follow human instructions. Despite their impressive performance in many tasks, they still face reliability issues such as hallucinations, stemming from misalignment with human instructions (Thakur et al., 2024; Ouyang et al., 2022; Ye et al., 2023; Wang et al., 2023; Zhou et al., 2023) or different modality information (Zhou et al., 2024a; Wang et al., 2024b; Yu et al., 2024b). To address these misalignment issues, recent studies have employed preference learning techniques—such as reinforcement learning from human feedback (RLHF) (Yu et al., 2024a; Sun et al., 2023) and direct preference optimization (DPO) (Deng et al., 2024a; Rafailov et al., 2024), to align the outputs of foundation models with human preferences in LLMs or to harmonize multimodal knowledge in LVLMs.

The success of preference fine-tuning techniques hinges on the availability of high-quality, large-scale preference datasets. Researchers currently employ two main methods for constructing these datasets. The first involves human annotation, which yields high-quality data but is often limited in scale due to its labor-intensive nature (Yu et al., 2024a; Ji et al., 2024). The second method uses external AI models to generate preference data Li et al. (2023c); Zhou et al. (2024a); however, this approach may fail to capture the inherent preferences of the target model being fine-tuned, rendering the generated data less useful. Recently, the self-rewarding (Zhou et al., 2024a; Yuan et al., 2024; Wang et al., 2025) approach samples the target model's own outputs as responses and uses the model itself to reward these responses, constructing preference pairs. While promising, this

---

[*]Lead author, [†]Core contributor

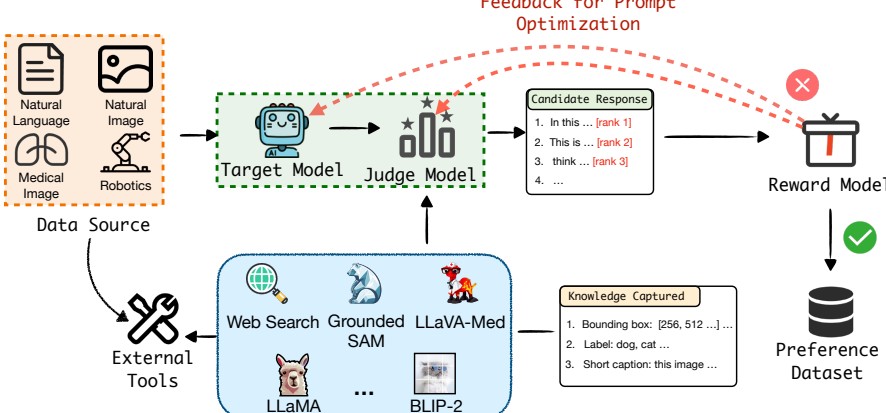

Figure 1: The figure illustrates the `Anyprefer` framework. First, `Anyprefer` selects the necessary tools based on the input prompt to obtain supplementary information, which is then integrated into a knowledge base. Next, the target model generates several responses for the input data. The judge model then ranks these responses using the constructed knowledge base. Subsequently, `Anyprefer` combines the best and worst-ranked responses into a preference pair. The reward model will then evaluate the quality of this preference pair, and all unqualified pairs will go through the optimization stage to refine its quality by using the proposed feedback mechanism.

method depends on the performance of the target model when serving as its own reward model. Inaccurate rewarding can bias the generated preference pairs, seriously compromising data quality. Therefore, improving the process of synthetic preference data synthesis is crucial for effective preference fine-tuning, given the scarcity of high-quality preference data and the challenges associated with annotation.

In this paper, as illustrated in Figure 1, we propose `Anyprefer`, a self-evolving synthetic preference data synthesis framework designed to automatically curate high-quality preference datasets. `Anyprefer` models the preference data synthesis process as a two-player cooperative Markov game between the *Target Model* and the *Judge Model*, parameterized by input prompts, to achieve a universal goal: maximizing the quality of preference data, as reflected by feedback from the *Reward Model*. Here, the goal for the *target model* is to generate high-quality pairwise preference data and the goal for the *judge model* is to provide robust and consistent ranking for the generated response. Achieving this universal goal requires collaboration between the target model and the judge model. `Anyprefer` supports various downstream applications, such as natural language generation, natural vision-language understanding, medical image analysis, and visuo-motor control. Specifically, `Anyprefer` generates preference data following the process of (1) response sampling, (2) response rewarding, (3) data quality evaluation, and (4) prompt optimization. First, in sampling stage, the *target model* generates a set of candidate responses based on the input prompts. Next, the *judge model* leverages external tools to gather relevant knowledge for rewarding these responses. Once ranked, the responses are used to construct preference data, which is then fed into a reward model to evaluate whether the preference data meets general quality criteria. Finally, with the feedback from the *reward model*, we refine the policy of the target model and the policy for the judge model by improving the prompt for these two models. Throughout this process, the target model and judge model act as cooperative players, working together to enhance preference data quality.

**Why Introducing Tools in Judge Model?** The inclusion of external tools is essential for ensuring annotation accuracy. `Anyprefer` strategically selects tools based on the input data to extract valuable information, mitigating bias during response rewarding. Additionally, the feedback mechanism introduced in the policy stage not only dynamically adjusts input prompts but also shares feedback with these tools, further enhancing their performance in supporting the judge model.

In summary, the primary contribution of this paper is `Anyprefer`, the first automatic framework for preference data synthesis. Experimental results across four key applications—natural language generation, vision-language understanding, medical image analysis, and visuo-motor control—spanning 21 datasets or tasks, demonstrate the effectiveness and advantages of `Anyprefer` in generating high-quality preference data and facilitating effective preference fine-tuning. In these four applica-

Table 1: Statistics comparison of `Anyprefer-V1` with existing preference datasets. The column "Scale" stands for the size of the generated dataset. In the column "Applications", `NL` stands for natural language tasks, `IMG` stands for natural images tasks, `MED` stands for medical tasks and `CTRL` stands for visuo-motor control tasks. In the column "Data Type", Img-Txt stands for image-text, Img-Ctrl-Seq stands for image-control sequences. Column "Multi-iter" stands for if the generation process is a multi-iteration process or not.

| Dataset Name | Scale | Human Effort | Response Generator | Tasks | Data Type | Multi-iter. |
|---|---|---|---|---|---|---|
| HH-RLHF | 161K | High | Human Label | `NL` | Text | No |
| Nectar | 183K | Low | GPT-4 | `NL` | Text | No |
| Orca-DPO-Pairs | 13K | Low | GPT-4 | `NL` | Text | No |
| UltraFeedback | 64K | Low | GPT-4 | `NL` | Text | No |
| LLaVA-RLHF | 10K | High | Llava | `IMG` | Img-Txt | No |
| RLAIF-V | 34K | Low | MLLM | `IMG` | Img-Txt | No |
| POVID | 17K | Low | GPT-4+Target Model | `IMG` | Img-Txt | No |
| VLFeedback | 80K | No | Open source LVLMs | `IMG\|MED` | Img-Txt | No |
| `Anyprefer-V1` | 58K | No | Target model | `NL\|IMG\|`
`MED\|CTRL` | Text; Img-Txt;
Img-Ctrl-Seq | Yes |

tions, `Anyprefer` achieves improvements of 18.55%, 3.66%, 30.05%, and 14.50%, respectively. Additionally, our experiments demonstrate the effectiveness of the tool-augmented judgment and feedback mechanism. Furthermore, we have compiled the synthesized data into a new preference dataset, `Anyprefer-V1`, comprising 58K high-quality preference pairs. As shown in Table 1, compared to previous synthesized preference data, `Anyprefer-V1` includes a broader range of application scenarios and data types. This will benefit the open-source community and further advance AI alignment research.

## 2 ANYPREFER

To address the challenges of synthesizing high-quality preference data, we propose an automatic framework called `Anyprefer`, which models the preference data synthesis process as a two-player cooperative Markov game. As illustrated in Figure 1, the target model and the judge model serve as two collaborative players working together to perform preference data synthesis. The target model first generates response candidates based on the input prompt, while the judge model integrates information from various tools to accurately reward and rank the responses. The ranked candidates are then evaluated by a reward model to ensure they meet general data quality criteria. Feedback from the reward model is used to optimize both the input prompts and the tools employed, enhancing the quality of low-quality preference data pairs. Ultimately, qualified preference pairs are used as preference data for preference fine-tuning. In the following sections, we will first detail the problem formulation and then discuss how to generate the preference data for preference fine-tuning.

### 2.1 PROBLEM FORMULATION

We discuss the formulation of the proposed `Anyprefer` framework. To begin with, we denote the input data prompt as $\mathbf{x}$ (e.g., a natural image) and the set of knowledge tools $\{\mathcal{M}_i\}_{i=1}^M$. Each knowledge tool $\mathcal{M}_i$ (e.g., Grounded SAM (Ren et al., 2024)) takes the data $\mathbf{x}$ as the input and output a sequence $\mathbf{q}_i = \mathcal{M}_i(\mathbf{x})$ extracting the information from $\mathbf{x}$ using model $\mathcal{M}_i$ as a delegate.

We model the preference data synthesis as a two-player cooperative Markov Game (MG). In particular, the first player is the target model $\pi_t$ which takes the data $\mathbf{x}$ as input and generate a set of candidates $\{\mathbf{y}_c\}_{c=1}^C$. The second player is the judge model $\pi_j$, it takes the candidate set $\{\mathbf{y}_c\}_{c=1}^C$ and the knowledge base model $\{\mathbf{q}_i\}_{i=1}^M$ as an input, then outputs the preference pair $\{\mathbf{y}_+, \mathbf{y}_-\}$. From the model selection perceptive, judge model $\pi_j$ actively aggregates the information from $\mathbf{q}_i$ and rank the $\{\mathbf{y}_c\}$ output by $\pi_t$. Since both $\pi_t$ and $\pi_j$ are language-based models, the input prompt $\mathbf{p}_t$ and $\mathbf{p}_j$ can be used to serve as their parameters, respectively. The goal of this MG is to generate a set of preference pair $\{\mathbf{y}_+, \mathbf{y}_-\}$ by the collaboration between the judge model and the target model, so that the collected preference data can improve the preference fine-tuning of the target model $\pi_t$. Generally, it is costly and time-consuming to directly evaluate the preference fine-tuning performance in

every step, we instead use a reward model $\mathcal{R}(\mathbf{y}_+, \mathbf{y}_-)$ to provide a surrogate reward by evaluating whether the target model benefits from the preference data $\{\mathbf{y}_+, \mathbf{y}_-\}$. Therefore the goal of this framework can be formulated as

$$\underset{\mathbf{p}_t, \mathbf{p}_j}{\arg\max} \, \mathbb{E}_{(\mathbf{y}_+, \mathbf{y}_-)} \big[ \mathcal{R}(\mathbf{y}_+, \mathbf{y}_-) \mid \pi_t(\cdot | \mathbf{p}_t), \pi_j(\cdot | \mathbf{p}_j), \mathbf{x}, \{\mathbf{q}_i\}_i \big], \tag{1}$$

where the expectation is taken over $(\mathbf{y}_+, \mathbf{y}_-) \sim \pi_j(\cdot | \{\mathbf{y}_c\}_c; \{\mathbf{q}_i\}_i; \mathbf{p}_j)$ and $\mathbf{y}_c \sim \pi_t(\cdot | \mathbf{x}; \mathbf{p}_t)$. According to equation 1, in the preference data generation process, it is feasible to optimize prompt $\mathbf{p}_t$ and $\mathbf{p}_j$ simultaneously using policy optimization with prompt-based gradient ascent (Pryzant et al., 2023). We provide a more detailed discussion and additional results in Appendix B to highlight the significance of the two-play cooperation framework.

## 2.2 Response Sampling and Rewarding

To synthesize preference data using `Anyprefer`, the first stage is sampling several candidate responses. Specifically, for a given input prompt $\mathbf{x}$, we sample $C$ unique response candidates $\{\mathbf{y}_c\}_{c=1}^C$ from the target model $\pi_t(\cdot | \mathbf{p}_t)$, where $\mathbf{p}_t$ is initialized with the input prompt $\mathbf{x}$. In our experimental setup, $C$ is universally set to 5, balancing diversity of samples with sampling costs.

After sampling the candidate responses, the next step is to use the judge model to accurately reward and rank these responses $\{\mathbf{y}_c\}_{c=1}^C$. To reduce potential bias from relying solely on the target model for evaluation (Yuan et al., 2024; Guo et al., 2024), we introduce a tool-augmented rewarding strategy for a more comprehensive evaluation. These knowledge tools gather relevant information from various perspectives to assist the judge model $\pi_j$ in providing accurate rewards. Based on the input prompt and candidate response, along with its own parameters (policy), i.e., the system prompt $\mathbf{p}_j$, the judge model strategically aggregates information captured by external tools for evaluation. Specifically, the tools extract relevant information $\mathbf{q}_i = \mathcal{M}_i(\mathbf{x})$ from the input prompt $\mathbf{x}$. The judge model $\pi_j$ then leverages this extracted knowledge $\mathbf{q}_i$ to provide an overall score $\pi_j(\cdot | \mathbf{y}_c; \{\mathbf{q}_i\}_i; \mathbf{p}_j)$ for each candidate response $\mathbf{y}_c$. Finally, the candidates are ranked, and the top-scoring response is selected as the preferred response $\mathbf{y}_+$, while the lowest-scoring is selected as the dispreferred response $\mathbf{y}_-$, forming the preference pair $\{\mathbf{y}_+, \mathbf{y}_-\}$. The initial system prompt $\mathbf{p}_j$ used in the judge model are detailed in Appendix D. And note that this prompt as part of the policy parameters can be constantly updated through the formulated two-player MG framework.

## 2.3 Data Quality Evaluation

Ideally, after identifying the preference pair $\{\mathbf{y}_+, \mathbf{y}_-\}$, we can directly use it to fine-tune the target model, collecting performance feedback to enhance the prompts $\mathbf{p}_j$ and $\mathbf{p}_t$ of both the judge model and target model. This, in turn, improves the data synthesis process. However, the fine-tuning process can be costly and time-consuming, which prevents the immediate feedback for updating the judge model and the target model, setting barriers for effectively optimizing the policy. To address this issue, we instead adapt LLM-as-a-Judge strategy (Zheng et al., 2023) to a LLM-based reward model $\mathcal{R}$ to judge the data quality. Here, the used LLM-as-a-Judge prompt can be found in the Appendix D. This reward model can evaluate the quality of the generated preference pair $\{\mathbf{y}_+, \mathbf{y}_-\}$ and return a reward $\mathcal{R}(\mathbf{y}_+, \mathbf{y}_-)$ that reflects the quality, and diversity of every preference pair. Generated preference pairs with high-quality rewards will be directly collected into the final preference dataset, while the others will be re-generated via the cooperation between the target model and judge model, using an updated policy guided by the reward $\mathcal{R}(\mathbf{y}_+, \mathbf{y}_-)$.

## 2.4 Learning from the Feedback

To effectively refine and improve the filtered low-quality preference data, we can use the obtained reward $\mathcal{R}(\mathbf{y}_+, \mathbf{y}_-)$ as the feedback to optimize the policy of the target model and judge model as illustrated in equation 1. Specifically, for updating the policy of the target model $\pi_t$, the input prompt $\mathbf{p}_t$ can be optimized to increase the probability of sampling more high-quality and diverse responses from the target model $\pi_t$. For updating the policy of the judge model $\pi_j$, the used system prompt $\mathbf{p}_j$ will be also optimized, which will finally affect the aggregation of the tools information. Motivated by Pryzant et al. (2023) and Yuksekgonul et al. (2024), this policy optimization process is similar to normal gradient descent, where the feedback $\mathcal{R}(\mathbf{y}_+, \mathbf{y}_-)$ can be viewed as the gradients passing through the models to update their parameters $\mathbf{p}_t$ and $\mathbf{p}_j$. Thus, we formulae this process as follows:

$$\mathbf{p}_t \leftarrow \mathbf{p}_t + \eta \nabla_{\mathbf{p}_t} \mathbb{E} \big[ \mathcal{R}(\mathbf{y}_+, \mathbf{y}_-) \big], \qquad \mathbf{p}_j \leftarrow \mathbf{p}_j + \eta \nabla_{\mathbf{p}_j} \mathbb{E} \big[ \mathcal{R}(\mathbf{y}_+, \mathbf{y}_-) \big], \tag{2}$$

---

**Algorithm 1** Anyprefer Framework for Preference Data Synthesis

---

**Require:** Dataset $\mathcal{D}$; Target model $\pi_t$; Judge model $\pi_j$; Reward model $\mathcal{R}$; Knowledge tools $\{\mathcal{M}_i\}_{i=1}^M$; Reward threshold $\tau$

**Ensure:** A set of high-quality preference pairs and optimized prompts $\mathbf{p}_t$, $\mathbf{p}_j$

    **for** each $x \in D$ **do**
        **repeat**
            **1.** Generate candidate responses $\{\mathbf{y}_c\}_{c=1}^C$ using the target model $\pi_t$ with prompt $\mathbf{p}_t$
            **2.** $\pi_j$ aggregates knowledge $\{\mathbf{q}_i\}_{i \in \mathcal{S}}$ from external tools $\{\mathcal{M}_i\}_{i \in \mathcal{S}}$ for each candidate response $y_c$,
where $\mathcal{S}$ is the selected tools decided by the strategy of $\pi_j$
            **3.** Compute judge scores $\pi_j(\cdot|\mathbf{y}_c; \{\mathbf{q}_i\}_{i \in \mathcal{S}}; \mathbf{p}_j)$ for each candidate response $y_c$ using the judge model
$\pi_j$ with knowledge $\{\mathbf{q}_i\}_{i \in \mathcal{S}}$
            **4.** Rank candidate responses $\{\mathbf{y}_c\}_{c=1}^C$ based on judge scores
            **5.** Select top-scoring and lowest-scoring responses to form preference pairs $(\mathbf{y}_+, \mathbf{y}_-)$
            **6.** Evaluate preference pairs using $\mathcal{R}$ to obtain reward $\mathcal{R}(\mathbf{y}_+, \mathbf{y}_-)$
            **if** $\mathcal{R}(\mathbf{y}_+, \mathbf{y}_-) < \tau$ **then**
                Update prompts $\mathbf{p}_t$ and $\mathbf{p}_j$ using policy gradient ascent based on $\mathcal{R}(\mathbf{y}_+, \mathbf{y}_-)$
        **until** $\mathcal{R}(\mathbf{y}_+, \mathbf{y}_-) \geq \tau$

---

where $\eta$ is the prompt adjustment step. The above policy gradient method aims at iteratively refining the input prompt (parameters) $\mathbf{p}_t$ and $\mathbf{p}_j$ of the target model $\pi_t$ and judge model $\pi_j$, respectively. By iteratively updating these parameters, the updated players $\{\pi_t, \pi_j\}$ are expected to better cooperate on generating preference pairs that meet criteria of the reward model and increase the reward. Finally, the proposed policy optimization are expected to effectively enhance the quality of the generated preference data.

## 2.5 ITERATIVE PREFERENCE FINE-TUNING

In this section, after curating the high-quality preference data, we fine-tune the target model through Direct Preference Optimization (DPO) (Rafailov et al., 2024). This process yields a stronger model, which we then replace as the target model. Then, the enhanced target model collaborates with the judge model to generate new preference data, which is subsequently used to fine-tune the target model. This iterative process can be repeated for multiple rounds. Details are in Appendix A.1.

## 3 EXPERIMENT

In this section, empirically demonstrate how the preference data constructed by `Anyprefer` effectively enhances the performance of various foundation models across four downstream applications. We address the following key questions: (1) Does the preference data generated by `Anyprefer` improve model performance across diverse applications and benchmarks? (2) Can `Anyprefer` boost the capabilities of different foundation models through iterative preference learning? (3) Is there a positive correlation between the surrogate reward provided by the reward model and the performance of preference fine-tuning on the target model (i.e., the actual reward)? (4) What is the quality of the preference data automatically synthesized by `Anyprefer`?

## 3.1 APPLICATIONS AND EXPERIMENTAL SETUPS

This section provides an overview of the downstream applications along with their corresponding experimental settings, deployment details, evaluation benchmarks, and baselines. The downstream applications include natural language generation, vision-language understanding, medical image analysis, and visuo-motor control, which are detailed below:

**Natural Language Generation.** The first application is using large language models for natural language generation. We utilize LLaMA2-7B-chat (Touvron et al., 2023) as the target model. We use GPT-4o as the judge model, which will utilize two tools: DuckDuckGo for web search (duc) and `FsfairX-LLaMA3-RM-v0.1` (Xiong et al., 2024) for response quality assessment. The GPT-4o is also adopted as the reward model to provide the immediate feedback for the generated preference pair. For baseline methods, we include original LLaMA2 model, self-rewarding Yuan et al. (2024) and meta rewarding Wu et al. (2024a) for comparison. For evaluation, we use three natural language benchmarks: GSM8K (Cobbe et al., 2021), ARC-easy/challenge (Clark et al., 2018), and AlpacaEval (Li et al., 2023d), covering commonsense question answering, math reasoning and alignment domains. Implementation details are provided in Appendix A.2.

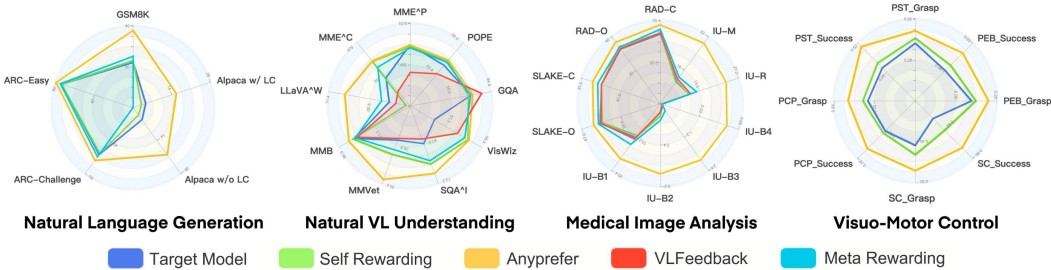

Figure 2: We evaluated `Anyprefer` using benchmarks from four applications. The target model represents the original model before preference fine-tuning. For medical image analysis, "B" for BLEU, "R" for ROUGE-L, "M" for METEOR, "C" for closed, and "O" for open tasks. In medical iamge analysis, "RAD": VQA-RAD, "IU": IU-Xray.

**Natural Vision-Language Understanding.** The second downstream application is using large Vision-Language Models (LVLMs) for natural vision-language understanding. In this application, we use LLaVA-1.5 7B as the target model. For tool selection, we leverage several state-of-the-art vision models as external knowledge sources, including the visual detection model Florence-2-large (Xiao et al., 2023), the short captioning model BLIP-2 (Li et al., 2023b), and the detection and segmentation model Grounded SAM (Ren et al., 2024). Additionally, we employ a powerful central multimodal model, GPT-4o, to integrate and interpret all the information for judgment and reward assessment. For baselines, we compare original LLaVA-1.5 7B model and LLaVA-1.5 7B with the self-rewarding approach Yuan et al. (2024), vlfeedback Li et al. (2024a) and meta rewarding Wu et al. (2024a). For evaluation, we follow the setup from Zhou et al. (2024a) and validate `Anyprefer` on three types of benchmarks: comprehensive benchmarks, general QA benchmarks, and hallucination benchmarks. For specific configurations, please refer to Appendix A.3.

**Medical Image Analysis.** Furthermore, we also evaluate `Anyprefer` in medical image analysis (MIA). Here, we use LLAVA-Med v1.5 (Li et al., 2023a) as the target model, which is a variant of LLaVA fine-tuned specifically for medical image understanding. For the tools and reward model selection, we use several powerful medical models in specific tasks (e.g., detection, captioning) as external knowledge source, including MiniGPT-Med (Alkhaldi et al., 2024), MedVInT (Zhang et al., 2023), CheXagent (Chen et al., 2024c) and a powerful central multimodal model (i.e., GPT-4o) for understanding and integrating all the information into judgment and rewarding. It is worthwhile to noting that the current Med-LVLMs are unable to generate high-quality data as preferred responses (Xia et al., 2024). Therefore, unlike natural language generation and vision-language understanding applications, we utilize the target model solely to synthesize dispreferred responses (Chen et al., 2024d), while the ground truth serves as the preferred responses. For evaluation, we conduct experiments on two tasks using three datasets: VQA-RAD (Lau et al., 2018) and SLAKE (Liu et al., 2021) for the medical VQA task, and IU-Xray (Demner-Fushman et al., 2016) for the report generation task. Implementation details are provided in Appendix A.4.

**Visuo-Motor Control.** The final application in `Anyprefer` is using vision-language-action model for visuo-motor control (VMC). In this case, we employ OpenVLA (Kim et al., 2024) as the target model. To implement `Anyprefer`, we use the image segmentation model Grounded SAM 2 (Ren et al., 2024) as a tool to segment the objects involved in the tasks and obtain their pixel coordinates. We then employ GPT-4o as a judge model to generate trajectory cost functions based on the pixel coordinate information and task prompts, including path cost, grasp cost, and collision cost. Following a feedback mechanism, the feedback generated by the scoring model is fed back to the judge model to produce prompts better suited for the current task, improving object segmentation and trajectory generation through multiple iterations. For baselines, we include several mainstream robotic models, including RT-1 (Brohan et al., 2022), Octo-small (Team et al., 2024), Octo-base (Team et al., 2024), and OpenVLA-SFT (OpenVLA fine-tuned on the Simpler-Env (Li et al., 2024b) dataset through SFT). We evaluate our model and the baseline models on four WidowX Robots tasks within the Simpler-Env (Li et al., 2024b): "placing the carrot on a plate", "putting the spoon on a towel", "stacking the green cube on top of the yellow cube", and "placing the eggplant into a basket". We compare the generated trajectories with the ground truth trajectories, evaluating the accuracy of task completion by the generated trajectories. See detailed implementations in Appendix A.5.

## 3.2 MAIN RESULTS

In Figure 2, we compare `Anyprefer` with four key baselines: the original target model, self-rewarding, meta-rewarding and vlfeedback. Detailed results, along with values from additional baselines tailored to each specific application, are provided in Table 4 to 15 in Appendix. Overall, `Anyprefer` demonstrates significant improvements across various applications, including natural language generation, vision-language understanding, medical image analysis, and visuomotor control. Specifically, in natural language generation, `Anyprefer` achieves up to a 10.92% increase in accuracy on the GSM8K and ARC datasets compared to the best baseline. On vision-language understanding benchmarks, `Anyprefer` outperforms both the original LLaVA-1.5 and the self-rewarding approach, notably achieving a 6.8% improvement on the VisWiz dataset. For medical image analysis, `Anyprefer` delivers the best performance, with an average improvement of 31.05% in medical VQA and report generation tasks. In visuomotor control, we observed success rate increases of up to 14.5% across various tasks.

Additionally, the self-rewarding approach and meta-rewarding also surpass the original target model, further demonstrating the effectiveness of synthesized preference data. By integrating tool information and feedback-guided policy optimization, `Anyprefer` significantly enhances the model's ability to generate more accurate and high-quality responses, making the constructed preference data more precise and effective. Moreover, in specialized domains like medical image analysis and visuomotor control, where data scarcity often leads to unstable performance in target models, the inclusion of additional tools and feedback mechanisms helps overcome the knowledge limitations of the original models, resulting in substantial performance gains.

## 3.3 ABLATION STUDY

We conduct ablation studies to evaluate the effectiveness of incorporating tools for response judgment and the feedback mechanism for policy optimization. The results in Table 2 demonstrate that introducing additional tools significantly improves overall model performance compared to the original model that only use GPT-4o as the judge model. This outcome aligns with our expectations, as the external tools enhance the comprehensiveness of the judge model in rewarding and ranking candidate responses, while also reducing bias in the ranking pro-

Table 2: Ablation study on the impact of tools and feedback. The table presents the average scores for each benchmark. "T" represents tool-augmented judgment, and "F" represents feedback mechanism.

| T | TF | LLM | LVLM | Med-LVLM | VLA |
|---|---|---|---|---|---|
| | | 56.88 | 67.90 | 23.35 | 28.0 |
| ✓ | | 59.88 | 68.82 | 25.24 | 30.5 |
| ✓ | ✓ | **61.03** | **69.61** | **30.60** | **40.5** |

cess to some extent. Moreover, incorporating the feedback mechanism to optimize the policy—both the prompts for the target model and the judge model—further boosts performance, with an average improvement of 21.51% across all applications. For more specific results, please refer to Tables 5, 9, 12 and 15 in the Appendix. These findings indicate that the feedback mechanism elevates the quality of preference data, thereby strengthening the target model. To further validate the role of tools in `Anyprefer` and the benefits of the joint two-player framework, we have conducted detailed ablation studies on these two aspects in Appendix B, specifically in Tables 5, 9, 15, and 12.

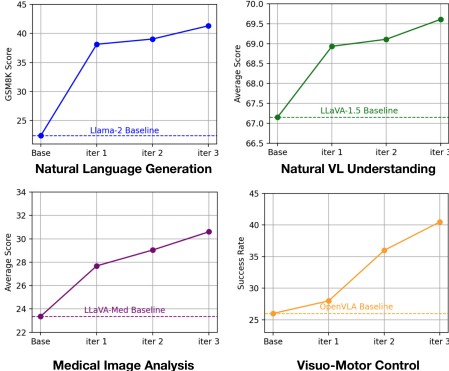

Figure 3: The performance of `Anyprefer` at different iterations over all applications.

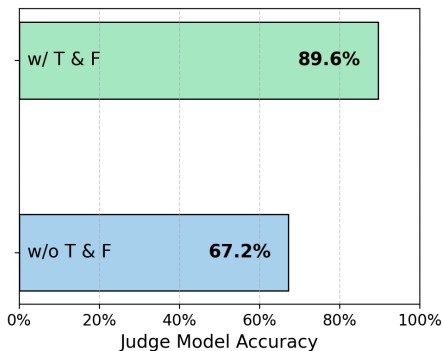

Figure 4: Impact of tools (T) and feedback (F) on judge model.

### 3.4 CAN ANYPREFER SUPPORT MODEL SELF-IMPROVEMENT?

In this section, we validate if `Anyprefer` can continuously improve model performance across four applications through iterative updates. At each iteration, the `Anyprefer` framework generate the preference data, and then use the data to fine-tune the target model. As shown in Figure 3, we report the performance of `Anyprefer` in natural language generation, vision-language understanding, medical image analysis, and visuomotor control. Through multiple iterative updates, `Anyprefer` exhibits significant performance improvements in all tasks. For instance, in natural language generation, the model demonstrates a notable score increase on the GSM8K dataset compared to the baseline. Similarly, in vision-language understanding and medical image analysis, the model demonstrates significant progress, achieving improvements of 3.66% and 31.02%, respectively. In the visuo-motor control task, `Anyprefer` shows the most significant improvement in success rate, with a 16.00% increase compared to the base model. These results indicate that `Anyprefer` exhibits strong self-improvement capabilities across all four applications, improving the quality of preference data with each iteration, leading to better overall model performance.

### 3.5 ANALYSIS OF JUDGE MODEL

In this section, we use natural vision-language understanding as an example to analyze the scoring accuracy of the judge model with and without tools (T) and feedback mechanism (F). We manually selected 200 examples, consisting of 100 samples generated using tool-captured knowledge and feedback mechanisms, and 100 samples generated without them. A human evaluation was conducted following the criteria outlined in Appendix D. The results, as shown in Figure 4, demonstrate that the introduction of tools and feedback mechanisms significantly improves the accuracy of the judge model: with tools and feedback mechanisms, the judge model's accuracy reaches 89.6%, whereas without them, it is only 67.2%, showing an absolute improvement of approximately 22.4%. This suggests that tools and feedback mechanisms can greatly enhance the judge model's evaluation accuracy, resulting in better ranking of responses generated by the target model.

### 3.6 ANALYSIS OF REWARD MODEL

Furthermore, we conducted experiments to evaluate whether the surrogate reward scores provided by the reward model in `Anyprefer` are highly correlated with the actual reward scores, i.e., the preference fine-tuning performance of the target model. We compared the correlation between the target model's performance over three preference fine-tuning iterations in `Anyprefer` and the surrogate reward scores corresponding to the preference data pairs generated by the target model during those iterations. As shown in Figure 5, the preference data produced by `Anyprefer` consistently improves the target model's performance across all four applications over three iterations. Moreover, as the iterations progress, the average surrogate reward score generated by our reward model increases in parallel with the target model's performance. This indicates a strong correlation between the

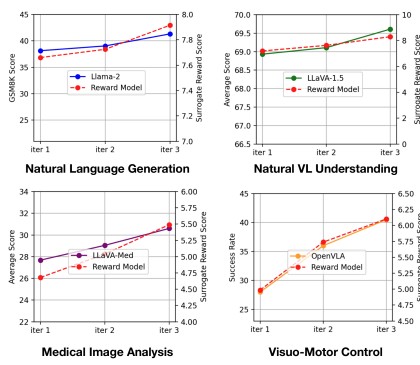

Figure 5: Impact of tools and feedback on judge model accuracy.

surrogate reward scores and the direct evaluation results of preference tuning, demonstrating the effectiveness of our reward model in providing reliable surrogate rewards.

### 3.7 ANALYSIS OF SYNTHESIZED DATASET DIVERSITY AND QUALITY

In this section, we evaluate the preference data `Anyprefer-V1` synthesized by `Anyprefer`, comparing it against existing synthesized preference datasets to verify its diversity and qualtiy. Diversity is analyzed using methods from (Zhao et al., 2024), while data quality are evaluated through manual annotations and GPT-4 scoring, which are detailed as follow:

**Data Diversity.** For diversity, we categorize the datasets in Table 1 into two groups: natural language datasets and multimodal datasets. We select two representative datasets from each group and randomly sample 2,000 instances from each. Specifically, HH-RLHF and Orca are chosen for the natural language group, while LLaVA-RLHF and VLFeedback are selected for the multimodal group. The text data from both groups are mapped using the text encoder

from CLIP-ViT-Base, and the image data in the multimodal group are mapped using the target model's image encoder. We apply t-SNE (Van der Maaten & Hinton, 2008) to project these embeddings into a two-dimensional space, as shown in Figure 6 (see more quantitative analysis in Appendix C). The results show that `Anyprefer-V1` nearly covers the full range of other datasets, both for text-only and multimodal data. Moreover, it occupies regions of the embedding space that are not covered by other datasets, highlighting its greater diversity.

**Data Quality.** For quality assessment, we randomly sampled 800 examples for manual evaluation, focusing primarily on two aspects: the difficulty of the data and the satisfaction level with the data. Specific scoring criteria and guidelines are provided in Appendix D.3. The results, shown in Figure 7, demonstrate that the difficulty of the preference data constructed by our framework mostly falls within the moderate range, with a reasonable distribution that avoids being too difficult or too simple. Moreover, the human evaluation results indicate that annotators are generally

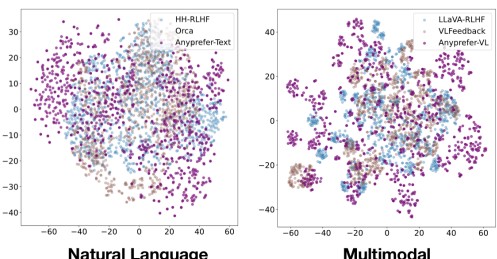

Figure 6: Comparison of `Anyprefer-V1` and other representative datasets in t-SNE mapping.

satisfied with the data generated by `Anyprefer`, which suggests that the preference data constructed by `Anyprefer` is of high quality. Furthermore, we randomly selected 200 examples from the VLFeedback, Orca, and our constructed `Anyprefer-V1` datasets, and used GPT-4o to score them on a scale of 1 to 10, with a higher score indicating higher data quality. The results are represented as bar charts in part (b) of Figure 7. From the results we can see that it is clear that the data constructed by our framework received relatively higher scores, aligning with the manual validation results. This further demonstrates the high quality of the data generated by `Anyprefer`.

## 3.8 CASE STUDY

In this section, we present and analyze several cases from the dataset, `Anyprefer-V1`, constructed by `Anyprefer`. We generated four cases, each corresponding to one application scenario: natural language generation, vision-language understanding, medical image analysis, and visuo-motor control, as shown in Figure 8. From the figure, we observe that the differences between the preferred and dispreferred responses in the preference pairs generated by `Anyprefer` are often quite subtle. For instance, in the vision-language understanding case, the dispreferred response mentions "kiwis and grapefruit," a minor discrepancy. This aligns with our expectation that more similar answers make it harder for the target model to differentiate between them. Furthermore, even in domains where preference data is scarce in literature, such as visuo-motor control, `Anyprefer` generates high-quality preference pairs. In one example, the preferred response successfully places the eggplant on the plate, while the dispreferred response nearly grabs the eggplant but ultimately fails.

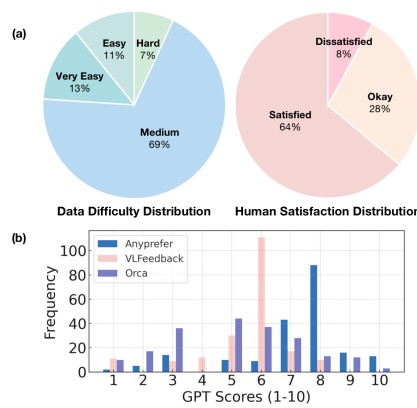

Figure 7: Data quality evaluation. (a) shows the results of manual evaluation from two aspects, and (b) represents the results of GPT-4o scoring.

## 4 RELATED WORK

Various empirical studies applying scaling laws (Kaplan et al., 2020; Hoffmann et al., 2022) to the training of foundation models have demonstrated the importance of the data size. To effectively scale the training data, synthetic data generation has emerged as a popular and cost-effective alternative, primarily leveraging advanced LLMs to produce high-quality data (Josifoski et al., 2023; Gunasekar et al., 2023; Taori et al., 2023; Chiang et al., 2023). In the post-training stage, especially for the preference training, high-quality preference data also faces the challenges in scaling.

**Preference Data Generation.** To effectively scale up the size of high quality preference data, self-play and self-rewarding methods have gained increasing attention as a practical method to self-

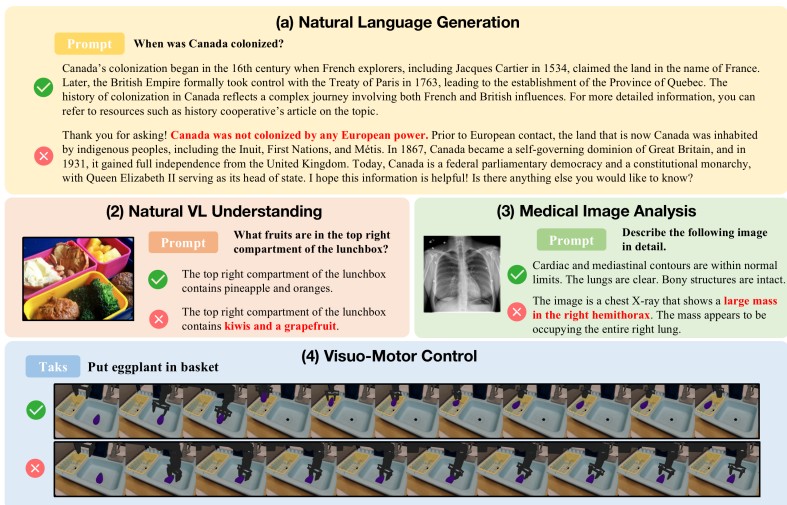

Figure 8: Case study. A checkmark indicates the preferred response, while a cross represents the dispreferred response. Errors and hallucinations in the dispreferred response are highlighted in red.

generate the training data without external supervision and models (Yuan et al., 2024; Singh et al., 2023; Chen et al., 2024d; Wu et al., 2024b; Cheng et al., 2024; Zhang et al., 2024). These methods usually consist of two steps: self-generating data and fine-tuning. And these two steps can be iteratively proceeding. Another line of research is Reinforcement Learning from AI Feedback (RLAIF) which utilizes advanced LLMs to label response pairs (Bai et al., 2022; Lee et al., 2023) for accurate rewarding and ranking. Meanwhile, the preference data generation for VLMs starts with CSR (Zhou et al., 2024b), which extends this concept to VLMs, to generate high quality vision-language preferences. Following CSR, SIMA (Wang et al., 2024b) is proposed to self-generate responses and employ an in-context self-critic mechanism to select response pairs for preference tuning. Similarly, Deng et al. (2024b) successfully applied the self-training manner to image comprehension.

Though these methods have successfully applied synthetic data generation to preference training, they commonly have the rewarding bias issue which means that their ranking annotations for those self-generated data are not accurate. For self-rewarding methods (Yuan et al., 2024; Singh et al., 2023; Chen et al., 2024d; Wu et al., 2024b; Cheng et al., 2024; Wang et al., 2024a; Chen et al., 2024b; Wang et al., 2025), there are no explicit constraints on the rewarding function, resulting in unreliable annotations. To mitigate this issue, our method introduces a series of external tools into the preference data rewarding process to ensure the rewarding accuracy. Existing works (Bai et al., 2022; Lee et al., 2023; Chen et al., 2024a; Tong et al., 2025) that use AI feedback to annotate preference data may alleviate the rewarding bias issue, however, they often overlook improving the quality of response sampling. To improve the quality of the sampled response, we introduce a two-player cooperative Markov Game framework to enable the immediate feedback for the policy model, which can help refine the response quality. Besides the proposed tool integration and feedback mechanism, we also apply preference data synthesis to multi domains including natural language generation, natural VL understanding, medical image analysis, and visuo-motor control, greatly benefiting the community.

## 5 CONCLUSION

This paper introduces the `Anyprefer` framework, an automatic system for synthesizing high-quality preference data across diverse applications. `Anyprefer` establishes a cooperative Markov game that synchronizes the target model with a judge model while incorporating external tools and feedback mechanisms. This approach improves both the quality and diversity of the generated preference data, `Anyprefer-V1`, which can be used to fine-tune the target model and enhance its performance. Experimental results show that the proposed `Anyprefer` framework significantly boosts performance in various applications such as natural language generation, vision-language understanding, medical image analysis, and visuo-motor control. Moreover, the experiments demonstrate the effectiveness of `Anyprefer` in enabling model self-improvement, and highlight the value of tool-augmented response judgment and feedback mechanisms.

ETHICS STATEMENT

This paper proposes the `Anyprefer` framework for automatically generating preference datasets, applied across multiple domains. The constructed datasets strictly adhere to ethical guidelines, ensuring that no sensitive information is included and minimizing potential bias during the data construction process. All experiments and data usage in this research comply with ethical standards. We acknowledge the potential issues related to fairness and bias that may arise when using automated tools for generating preference data. Therefore, we have adhered to relevant ethical standards throughout the data creation and evaluation process to ensure fairness and transparency. No personally identifiable information was collected or processed in this study.

REPRODUCIBILITY STATEMENT

To ensure the reproducibility of the results from `Anyprefer`, we provide detailed experimental setups and dataset construction processes. In Section 1, we explain the dataset creation, annotation guidelines, and data collection methods, with further elaboration in Appendix A. Additionally, in Section A, we provide a thorough description of the benchmark testing and evaluation procedures, with clearly defined metrics to facilitate independent verification of our results. To further support research and application in the community, we have also made the generated `Anyprefer` preference dataset publicly available for download and use by other researchers.

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

## A  EXPERIMENTAL SETUP

### A.1  TRAINING SETUP

For the training phase with preference data, after collecting each round of preference data, we use DPO to train for 3 epochs. The entire training process is conducted on a single A100 80G GPU. During training, we fine-tune the LoRA parameters for improved efficiency. Detailed training parameters can be found in Table 3.

Table 3: Training hyperparameters.

| Hyperparameters | |
| --- | --- |
| lora_r | 128 |
| lora_alpha | 256 |
| lora_target | all |
| mm_projector_lr | 2e-5 |
| Batch size | 1 |
| Learning rate | 1e-7 |
| model_max_length | 1024 |

### A.2  NATURAL LANGUAGE GENERATION

#### A.2.1  DATASET AND BASELINES

To evaluate our method, we use three datasets that target different model capabilities: (1) GSM8K (Cobbe et al., 2021) focuses on primary school-level math problems, requiring 2-8 steps of basic arithmetic to solve. We evaluate based on exact final answer matching. (2) ARC-easy/challenge (Clark et al., 2018) contains 7K grade-school science multiple-choice questions, split into an Easy Set and a Challenge Set (questions hard for both retrieval and word co-occurrence algorithms). We also use exact answer matching for evaluation. (3) AlpacaEval (Li et al., 2023d) tests general instruction-following, where model responses are compared to reference answers using GPT-4-based auto-annotators, with results reported as length controlled win rate (Dubois et al., 2024) and win rate.

As baselines, we include the untrained LLaMA2 model, a self-rewarding version of LLaMA2 following the methodology of Yuan et al. (2024), and an improved meta-rewarding Wu et al. (2024a) version of LLaMA2 with the addition of providing correct answers during the self-rewarding process when possible. Additionally, we conduct ablation studies by disabling the tools and feedback modules to evaluate their individual contributions to `Anyprefer`.

## A.3 Natural Vision-Language Understanding

### A.3.1 Dataset and Baselines

Besides the original LLaVA-1.5-7b model, its self-rewarding version and meta rewarding as baselines, we also incorporate a wide range of other preference data construct method, including: Silkie:(Li et al., 2023c) Constructs a VLFeedback dataset by generating responses from 12 LVLMs based on multimodal instructions. GPT-4V evaluates these responses on helpfulness, visual accuracy, and ethical considerations. LLaVA-RLHF: (Sun et al., 2023) Introduces Factually Augmented RLHF, an algorithm that improves the reward model by incorporating factual data such as image captions and ground-truth multi-choice answers. POVID: (Zhou et al., 2024a) Aligns VLLMs' preferences using external data from GPT-4 and the hallucination tendencies observed in noisy images. RLHF-V: (Yu et al., 2024a) Gathers human corrections on hallucinations at a paragraph level and applies dense direct preference optimization based on human feedback.

### A.3.2 Evaluation Benchmark

We conducted evaluations on three types of benchmarks: comprehensive benchmarks, general VQA and hallucination benchmarks. Specifically, this includes:

**MME:** (Fu et al., 2024) A broad benchmark for assessing LVLMs in multimodal tasks, focusing on both perception and cognition. It tests models across 14 subtasks that challenge their interpretative and analytical abilities.

**LLaVA$^W$:** (Liu et al., 2024) A visual reasoning benchmark with 24 diverse images and 60 questions, covering a range of scenarios from indoor or outdoor environments to abstract art.

**MMBench:** (Liu et al., 2023) Expands evaluation scope with a curated dataset and introduces the CircularEval strategy, which uses ChatGPT to transform free-form predictions into structured multiple-choice answers.

**MM-Vet:** (Yu et al., 2023) Assesses LVLMs through 16 multimodal tasks built from six core vision-language skills, providing detailed insights into model performance across various question types and response formats.

**ScienceQA:** (Saikh et al., 2022) A multimodal benchmark targeting multi-hop reasoning in science, containing 21K multiple-choice questions with associated explanations and lectures.

**VizWiz:** (Bigham et al., 2010) A VQA dataset with over 31,000 goal-oriented visual questions, featuring images taken by blind users and their spoken queries, along with crowdsourced answers.

**GQA:** (Hudson & Manning, 2019) A visual reasoning dataset with 22 million semantically-generated questions based on scene graphs, designed to evaluate consistency, grounding, and plausibility in model responses.

**POPE:** (Li et al., 2023e) A binary classification task to detect object hallucination in LVLMs, using yes or no questions and diverse object sampling strategies to expose hallucination tendencies.

## A.4 Medical Image Analysis

### A.4.1 Dataset and Baselines

We evaluate the performance of our method on three key datasets targeting medical image analysis tasks: (1) **VQA-RAD** (Lau et al., 2018) contains 3,515 question-answer pairs and 315 radiology images, with questions categorized into types like abnormality, modality, and organ system. Answers include both yes/no and open-ended responses. (2) **SLAKE** (Liu et al., 2021) consists of 642 radiology images and over 7,000 diverse QA pairs, requiring external medical knowledge and annotated with segmentation masks and bounding boxes. We only consider the English subset. (3) **IU-Xray** (Demner-Fushman et al., 2016) focuses on medical report generation, containing chest X-ray images paired with detailed clinical reports, evaluating the model's ability to generate accurate medical text based on images.

As baselines, we include the LLaVA-Med-1.5 model (Li et al., 2023a), a self-rewarding version of LLaVA-Med v1.5, and a meta-rewarding version of LLaVA-Med-1.5. Additionally, we adapt the VLFeedback method to LLaVA-Med-1.5 for comparison. Additionally, we perform ablation studies by disabling the tools and feedback modules to assess their individual contributions to `Anyprefer`.

### A.5 VISUO-MOTOR CONTROL

#### A.5.1 DATASET AND BASELINES

We employ Simpler-Env (Li et al., 2024b) as our experiment environment and dataset. SIMPLER (Simulated Manipulation Policy Evaluation for Real Robot Setups) is a suite of simulated environments designed to evaluate real-world robot manipulation policies. SIMPLER utilizes simulated environments as an effective proxy for real-world testing, addressing the challenges of real robot evaluations, which are typically expensive, slow, and difficult to reproduce.

To comprehensively assess the performance of our proposed method, we conducted baseline comparisons with several state-of-the-art robotic models. RT-1(Brohan et al., 2022) is a sophisticated robotic control system designed to handle real-world tasks at scale. It utilizes a Transformer-based architecture trained on approximately 130,000 demonstrations covering over 700 tasks, enabling it to generalize across a variety of tasks with minimal task-specific data. Octo(Team et al., 2024) is an open-source, generalist robot policy trained on 800,000 diverse robot episodes from the Open X-Embodiment dataset. Employing a transformer-based architecture, Octo demonstrates robust adaptation to various tasks, robots, and environments; we evaluated both its small (27M parameters) and base (93M parameters) versions. OpenVLA (Kim et al., 2024) is a 7B-parameter open-source vision-language-action model designed for generalist robot manipulation policies, trained on 970k robot demonstrations from the same dataset. Key features of OpenVLA include its ability to control multiple robots directly and its adaptability to new robot domains through efficient fine-tuning. We used the OpenVLA-baseline model, which was fine-tuned on the Simpler-Env dataset through supervised learning. These models were selected as baselines for comparison in our experiments to evaluate the effectiveness of our proposed method. Because OpenVLA can not generate word, use LLaVA-1.5-7B for self-rewarding. Regarding the dataset, the Simpler-Env dataset was created by using the OpenVLA model fine-tuned on the bridge-v2Walke et al. (2023) data to generate 500 successful trajectories within Simpler-EnvLi et al. (2024b).

#### A.5.2 EVALUATION BENCHMARKS

All the baseline models were tested on four WidowX robot tasks within the Simpler-Env:

1. Put the carrot on a plate
2. Put the spoon on a towel
3. Stack the green cube on the yellow cube
4. Put the eggplant in basket

For each task, we executed 50 trials where the positions of the source and target objects were randomly generated. The evaluation was based on whether the objects could be continuously grasped and whether the tasks were successfully completed. We compared the generated trajectories from each model with the ground truth trajectories, assessing their performance in terms of task success rate.

## B SUPPLEMENTARY EXPERIMENTS

### B.1 NATURAL LANGUAGE GENERATION

We present detailed results in Tables 4. `Anyprefer` achieves substantial improvements across all datasets, particularly when combined with external tools and feedback mechanisms. For natural language, on GSM8K and ARC datasets, our approach improves the absolute accuracy by 10.92%, 5.81% and 7.00% relative to the Pareto Optimal of untrained and self-rewarding baselines, clearly showcasing the strength of integrating external assistance. On AlpacaEval, our method outperforms

Table 4: Performance on text tasks. For GSM8K and ARC, we report the accuracy of the final answer. For Alpaca Eval, we report length controlled win rate / win rate (* indicates that the chosen response during the self-rewarding process uses the ground truth).

| Method | GSM8K | ARC-Easy | ARC-Challenge | Alpaca Eval 2.0 |
|---|---|---|---|---|
| Llama-2 | 22.44 | 74.33 | 57.68 | 5.20 / 4.57 |
| + Self Rewarding | 23.20 | 74.45 | 56.31 | 3.28 / 3.12 |
| + Self Rewarding* | 27.22 | 73.53 | 56.66 | - |
| + Meta Rewarding | 25.47 | 76.22 | 59.47 | - |
| + Anyprefer | **38.14** | **80.26** | **64.68** | **19.25 / 15.14** |

Table 5: Ablation study of natural language generation.

| Method | GSM8K | ARC-Easy | ARC-Challenge | Alpaca Eval 2.0 |
|---|---|---|---|---|
| *Feedback Mechanism Ablation* | | | | |
| Anyprefer | 30.10 | 78.16 | 62.37 | 3.99 / 3.75 |
| Anyprefer (tools) | 37.53 | 78.70 | 63.40 | 18.96 / 14.40 |
| Anyprefer (tools + feedback) | **38.14** | **80.26** | **64.68** | **19.25 / 15.14** |

Table 6: The multi-round preference iteration results of Llama2 and Anyprefer on the GSM8K dataset. The superscript "$l$" denotes LLaMA2, and the superscript "$a$" denotes Anyprefer (tools + feedback).

| Base$^l$ | Iter-1$^a$ | Iter-2$^a$ | Iter-3$^a$ |
|---|---|---|---|
| 22.44 | 38.14 | 39.04 | 41.32 |

Table 7: Comparison of different methods on natural vision-language understanding.

| Method | MME$^P$ | MME$^C$ | LLaVA$^W$ | MMB | MMVet | SQA$^I$ | VisWiz | GQA | POPE |
|---|---|---|---|---|---|---|---|---|---|
| LLaVA-1.5-7B | **1510.7** | 348.2 | 63.4 | 64.3 | 30.5 | 66.8 | 50.0 | 62.0 | 85.90 |
| + Vfeedback | 1432.7 | 321.8 | 62.1 | 64.0 | 31.2 | 66.2 | 52.6 | **63.2** | 83.72 |
| + Human-Prefer | 1490.6 | 335.0 | 63.7 | 63.4 | 31.1 | 65.8 | 51.7 | 61.3 | 81.50 |
| + POVID | 1452.8 | 325.3 | 68.7 | 64.9 | 31.8 | 68.8 | 53.6 | 61.7 | 86.90 |
| + RLHF-V | 1489.2 | 349.4 | 65.4 | 63.6 | 30.9 | 67.1 | **54.2** | 62.1 | 86.20 |
| + Self Rewarding | 1505.6 | 362.5 | 61.2 | 64.5 | 31.4 | 69.6 | 53.9 | 61.7 | 86.88 |
| + Meta Rewarding | 1498.3 | 357.4 | 64.0 | 64.2 | 31.3 | 69.1 | 53.5 | 62.0 | 86.70 |
| + Anyprefer | 1510.1 | **362.9** | **69.2** | **65.1** | **33.0** | **70.9** | 54.0 | 62.2 | **86.98** |

simpler setups with a more than threefold increase in win rates. In contrast, the self-rewarding mechanism alone struggles to deliver meaningful improvements, with gains being marginal at best. While self rewarding and meta rewarding offer some benefits, it alone cannot significantly enhance the performance of smaller models like LLaMA2-7B in complex tasks, indicating the need for additional support. Ablation studies further validate the effectiveness of each component in our approach. Disabling either the tools or feedback modules leads to notable performance declines, confirming that both elements are crucial to maximizing the model's potential.

## B.2 NATURAL VISION-LANGUAGE UNDERSTANDING

In this section, we present detailed experiment results on natural vision-language understanding.

Table 7 compares the performance of `Anyprefer` against other methods. The results demonstrate that `Anyprefer` consistently outperforms prior approaches across most benchmarks, highlighting the effectiveness of our framework and the robustness of the constructed dataset.

To further investigate the impact of key components within `Anyprefer`, we perform ablation studies by systematically removing the tool utilization feature and varying the feedback iterations. The outcomes of these studies are summarized in Table 9. Our findings reveal that integrating tools into

Table 8: The multi-round preference iteration results of LLaVA-1.5 on natural vision-language understanding.

| Method | $MME^P$ | $MME^C$ | $LLaVA^W$ | MMB | MMVet | $SQA^I$ | VisWiz | GQA | POPE |
|---|---|---|---|---|---|---|---|---|---|
| LLaVA-1.5-7B | **1510.7** | 348.2 | 63.4 | 64.3 | 30.5 | 66.8 | 50.0 | 62.0 | 85.90 |
| + Anyprefer Iter-1 | 1502.0 | 358.0 | 67.4 | 64.8 | 32.3 | 70.5 | 53.7 | 62.1 | 86.22 |
| + Anyprefer Iter-2 | 1506.5 | 360.3 | 67.2 | 64.9 | 32.4 | 70.7 | 53.6 | 62.0 | 86.95 |
| + Anyprefer Iter-3 | 1510.1 | **362.9** | **69.2** | **65.1** | **33.0** | **70.9** | **54.0** | **62.2** | **86.98** |

Table 9: Ablation study of natural vision-language understanding. For the rank ablation, we default to using lower-ranked responses as dispreferred data and higher-ranked responses as preferred data.

| Method | $MME^P$ | $MME^C$ | $LLaVA^W$ | MMB | MMVet | $SQA^I$ | VisWiz | GQA | POPE |
|---|---|---|---|---|---|---|---|---|---|
| *Feedback Mechanism Ablation* | | | | | | | | | |
| Anyprefer | 1488.5 | 340.4 | 64.3 | 64.7 | 31.7 | 69.9 | 53.4 | 62.0 | 86.92 |
| Anyprefer (tools) | 1498.2 | 357.5 | 66.8 | 64.6 | 32.1 | 70.3 | 53.6 | 62.1 | 86.90 |
| Anyprefer (tools + feedback) | **1510.1** | **362.9** | **69.2** | **65.1** | **33.0** | **70.9** | **54.0** | **62.2** | **86.98** |
| *Optimization Target Ablation* | | | | | | | | | |
| Optimize Target Model Only | 1480.2 | 350.2 | 65.2 | 63.9 | 31.0 | 67.2 | 52.0 | 61.5 | 86.10 |
| Optimize Judge Model Only | 1485.6 | 353.1 | 66.1 | 64.1 | 31.5 | 68.0 | 53.2 | 61.8 | 86.45 |
| Independently Optimize Both Models | 1495.8 | 359.0 | 67.8 | 64.7 | 32.5 | 69.2 | 53.5 | 62.0 | 86.70 |
| *Data Rank Ablation* | | | | | | | | | |
| Anyprefer (rank3 + rank5) | 1501.8 | 359.4 | 66.8 | 64.8 | 32.2 | 69.1 | 53.7 | 62.0 | 86.75 |
| Anyprefer (rank3 + rank1) | 1508.3 | 361.2 | 68.5 | 65.0 | 32.8 | 70.1 | 53.8 | 62.2 | 86.89 |

the framework enhances perceptual and cognitive capabilities, while increasing the number of feedback iterations yields additional performance gains. These results underscore the critical role that tools and feedback mechanisms play in our framework.

### B.3 MEDICAL IMAGE ANALYSIS

We evaluate the performance of models benefited from Anyprefer across two tasks and three widely-used datasets. As demonstrated in Figure 2, Anyprefer performs the best overall preformance, with an average improvement of 31.0%. As shown in Table 10, for medical VQA and report generation, the performance increased by 13.14% and 67.8%, respectively. Interestingly, we can also observe that model performance is improved significantly on report generation task, which is attributed to Anyprefer enhancing the open-ended generation capability. Compared with self-rewarding method, Anyprefer significantly outperforms the baseline method by 28.4%. By leveraging state-of-the-art medical models as external tools, we constructed an enhanced preference dataset, which significantly outperformed the self-rewarding approach. This improvement is attributed to the higher level of expertise and accuracy provided by specialized medical models in tasks such as VQA and medical report generation. Additionally, the integration of a powerful central multimodal model (e.g., GPT-4o) for information synthesis and reward judgment further enhances the model's ability to handle complex medical scenarios, resulting in significantly improved generation quality and accuracy.

Furthermore, the results indicate that increasing the number of external tools and incorporating feedback mechanisms both lead to notable improvements, particularly in medical report generation tasks. This suggests that our approach is especially effective for open-ended generation tasks. The improvement can be attributed to the enhanced capacity of the model to integrate domain-specific knowledge from multiple tools, while the feedback mechanism allows for iterative refinement, enabling the model to better capture the complexity and variability of medical reports, thereby producing more accurate and contextually appropriate outputs.

### B.4 VISUO-MOTOR CONTROL

The experimental results are presented in Table 13. Anyprefer , performed notably well compared to other models. With the integration of tools and feedback mechanisms, the performance across all tasks was further enhanced. The information provided by the tools improved the accuracy of

Table 10: Performance on medical VQA and report generation tasks. For open-set questions, we report the recall in column Open. For closed-set questions, we report the accuracy in column Closed. * indicates that the chosen response during the self-rewarding process uses the ground truth.

| | VQA-RAD | | SLAKE | | IU-Xray | | | | | |
| | Closed | Open | Closed | Open | BLEU-1 | BLEU-2 | BLEU-3 | BLEU-4 | ROUGE-L | METEOR |
|---|---|---|---|---|---|---|---|---|---|---|
| LLaVA-Med | 63.57 | 32.09 | 61.30 | 44.26 | 10.31 | 0.66 | 0.07 | 0.01 | 10.32 | 10.95 |
| + VLFeedback | 64.33 | 32.38 | 61.52 | 44.03 | 10.65 | 0.67 | 0.10 | 0.03 | 10.78 | 11.36 |
| + Self Rewarding | 64.17 | 33.29 | 61.30 | 42.63 | 9.71 | 0.97 | 0.10 | 0.01 | 10.38 | 10.52 |
| + Self Rewarding* | 66.25 | 32.19 | 63.28 | 42.80 | 9.56 | 1.03 | 0.18 | 0.02 | 11.14 | 11.83 |
| + Meta Rewarding | 67.42 | 33.05 | 65.10 | 45.12 | 12.48 | 1.23 | 0.24 | 0.03 | 12.56 | 13.21 |
| + Anyprefer | **72.06** | **36.10** | **70.39** | **49.04** | **16.85** | **5.57** | **2.07** | **0.56** | **23.69** | **29.66** |

Table 11: The multi-round preference iteration results of medical image analysis.

| | VQA-RAD | | SLAKE | | IU-Xray | | | | | |
| | Closed | Open | Closed | Open | BLEU-1 | BLEU-2 | BLEU-3 | BLEU-4 | ROUGE-L | METEOR |
|---|---|---|---|---|---|---|---|---|---|---|
| LLaVA-Med | 63.57 | 32.09 | 61.30 | 44.26 | 10.31 | 0.66 | 0.07 | 0.01 | 10.32 | 10.95 |
| + Anyprefer Iter-1 | 70.96 | 35.58 | 67.40 | 47.69 | 9.30 | 2.85 | 1.12 | 0.31 | 19.36 | 22.24 |
| + Anyprefer Iter-2 | 71.47 | 35.72 | 69.22 | 48.17 | 12.93 | 4.11 | 1.58 | 0.42 | 21.87 | 24.93 |
| + Anyprefer Iter-3 | **72.06** | **36.10** | **70.39** | **49.04** | **16.85** | **5.57** | **2.07** | **0.56** | **23.69** | **29.66** |

Table 12: Ablation study of medical image analysis. For the rank ablation, we default to using lower-ranked responses as dispreferred data and higher-ranked responses as preferred data.

| | VQA-RAD | | SLAKE | | IU-Xray | | | | | |
| | Closed | Open | Closed | Open | BLEU-1 | BLEU-2 | BLEU-3 | BLEU-4 | ROUGE-L | METEOR |
|---|---|---|---|---|---|---|---|---|---|---|
| *Feedback Mechanism Ablation* | | | | | | | | | | |
| Anyprefer | 66.73 | 32.14 | 64.66 | 44.75 | 9.30 | 1.19 | 0.32 | 0.05 | 12.42 | 15.72 |
| Anyprefer (tools) | 67.65 | 32.67 | 65.17 | 45.72 | 9.41 | 1.28 | 0.36 | 0.06 | 12.95 | 17.13 |
| Anyprefer (tools + feedback) | **72.06** | **36.10** | **70.39** | **49.04** | **16.85** | **5.57** | **2.07** | **0.56** | **23.69** | **29.66** |
| *Optimization Target Ablation* | | | | | | | | | | |
| Optimize Target Model Only | 66.20 | 33.53 | 65.36 | 45.79 | 12.89 | 3.16 | 1.20 | 0.28 | 14.87 | 17.99 |
| Optimize Judge Model Only | 68.71 | 34.39 | 67.48 | 46.60 | 13.37 | 3.79 | 1.42 | 0.36 | 17.89 | 19.60 |
| Independently Optimize Both Models | 70.21 | 34.89 | 68.37 | 47.65 | 14.57 | 4.18 | 1.68 | 0.43 | 20.66 | 23.74 |
| *Data Rank Ablation* | | | | | | | | | | |
| Anyprefer (rank3 + rank5) | 68.19 | 34.22 | 66.74 | 46.19 | 12.47 | 3.12 | 0.84 | 0.15 | 18.92 | 21.12 |
| Anyprefer (rank3 + rank1) | 70.15 | 35.41 | 68.32 | 47.83 | 14.58 | 4.37 | 1.56 | 0.32 | 20.53 | 25.47 |

the judge model, enabling the model to generate more accurate prompts and trajectories. From the comparison, it is evident that `Anyprefer` with tool and feedback mechanisms achieved the highest success rates on all tasks, significantly outperforming the other baseline models.

To evaluate the specific contributions of key components in our method to the overall performance, we conducted ablation experiments by removing the image segmentation model Grounded SAM (Ren et al., 2024) and the feedback mechanism. The experimental results are presented in Table 2 and Table 15

In the first ablation experiment, we assessed the performance of the model without using the image segmentation model Grounded SAM and feedback mechanism. This allowed us to understand the impact of the image segmentation model on object recognition and scene understanding.The experimental results showed that without Grounded SAM, the model's accuracy in locating and recognizing target objects significantly decreased, leading to an increased failure rate in trajectory generation. Specifically, the average success rate across the four tasks increased by approximately 15.51%.

In the second ablation experiment, we removed the feedback mechanism to observe how the absence of detailed feedback affects model training and trajectory generation.The experimental results indicated that without the feedback mechanism, the model struggled to optimize the generated trajectories, resulting in a lower success rate in task completion. The average success rate across the four tasks increased by approximately 17.46%, respectively.

As shown in Table 13 the integration of tools and feedback mechanisms led to relative improvements in the success rates of the four tasks by 26.67%, 38.89%, 38.46%, and 38.89%, respectively.

Table 13: Visuomotor-control: success rates for different tasks and models (* indicates that Open-VLA can not generate word, use LLaVA-1.5-7B as reward model).

| | Put Spoon on Towel | | Put Carrot on Plate | | Stack Cube | | Put Eggplant in Basket | |
| --- | --- | --- | --- | --- | --- | --- | --- | --- |
| | Grasp Spoon | Success | Grasp Carrot | Success | Grasp Cube | Success | Grasp Eggplant | Success |
| RT-1 | 0.10 | 0.06 | 0.20 | 0.12 | 0.22 | 0.02 | 0.06 | 0.00 |
| Octo-small | 0.42 | 0.28 | 0.30 | 0.16 | 0.42 | 0.10 | 0.48 | 0.32 |
| Octo-base | 0.38 | 0.20 | 0.22 | 0.10 | 0.24 | 0.04 | 0.46 | 0.32 |
| OpenVLA-SFT (baseline) | 0.52 | 0.28 | 0.36 | 0.30 | 0.56 | 0.20 | 0.58 | 0.32 |
| +Self Rewarding* | 0.54 | 0.28 | 0.38 | 0.30 | 0.56 | 0.22 | 0.54 | 0.34 |
| +Anyprefer | **0.60** | **0.38** | **0.72** | **0.50** | **0.70** | **0.36** | **0.84** | **0.50** |

Table 14: The multi-round preference iteration results of Visuomotor-control.

| | Put Spoon on Towel | | Put Carrot on Plate | | Stack Cube | | Put Eggplant in Basket | |
| --- | --- | --- | --- | --- | --- | --- | --- | --- |
| | Grasp Spoon | Success | Grasp Carrot | Success | Grasp Cube | Success | Grasp Eggplant | Success |
| OpenVLA | 0.52 | 0.28 | 0.36 | 0.30 | 0.56 | 0.20 | 0.58 | 0.32 |
| + Anyprefer Iter-1 | 0.52 | 0.30 | 0.46 | 0.36 | 0.52 | 0.26 | 0.70 | 0.36 |
| + Anyprefer Iter-2 | 0.58 | 0.34 | 0.66 | 0.46 | 0.62 | 0.34 | 0.80 | 0.46 |
| + Anyprefer Iter-3 | **0.60** | **0.38** | **0.72** | **0.50** | **0.70** | **0.36** | **0.84** | **0.50** |

Table 15: Ablation study of Visuomotor-control model.

| | Put Spoon on Towel | | Put Carrot on Plate | | Stack Cube | | Put Eggplant in Basket | |
| --- | --- | --- | --- | --- | --- | --- | --- | --- |
| | Grasp Spoon | Success | Grasp Carrot | Success | Grasp Cube | Success | Grasp Eggplant | Success |
| *Feedback Mechanism Ablation* | | | | | | | | |
| Anyprefer | 0.52 | 0.30 | 0.46 | 0.36 | 0.52 | 0.26 | 0.70 | 0.36 |
| Anyprefer (tools) | 0.56 | 0.34 | 0.60 | 0.42 | 0.60 | 0.30 | 0.80 | 0.42 |
| Anyprefer (tools+feedback) | **0.60** | **0.38** | **0.72** | **0.50** | **0.70** | **0.36** | **0.84** | **0.50** |

`Anyprefer` which combines tools and feedback, outperformed models that lacked either tools or feedback, and those with only tools.

## C   DIVERSITY EVALUATION

To further validate the diversity of the data, we selected the largest existing preference dataset, VLFeedback, as a baseline for comparison. A condition number-based approach was employed to evaluate the diversity of synthetic data (including VLFeedback and Anyprefer-v1). Specifically, we randomly sampled 500 examples from each dataset, constructed the covariance matrix of the data matrix (i.e., a sample-by-feature matrix), and calculated its condition number to quantify data diversity. A smaller condition number indicates a more dispersed distribution in the embedding space, reflecting higher diversity. As shown in Table 16, the condition number of the Anyprefer dataset is smaller, further supporting the conclusion that the Anyprefer dataset achieves higher diversity coverage.

## D   EVALUATION CRITERIA AND PROMPTS

In this section, we list the prompts used in `Anyprefer` and some of the rewarding criteria manually annotated during the experimental phase.

### D.1   JUDGE MODEL

**Judge Model Prompts**

**[Task]** Suppose that you are an expert in {`task_field`}, please rate the answers of some given questions.

Table 16: Comparison of condition numbers for VLFeedback and Anyprefer-v1 datasets.

| Method | Condition Number |
|---|---|
| VLFeedback | 1560.70 |
| Anyprefer-v1 | **1390.15** |

---

**[Guideline]** Focus on correctness (whether the information provided in the answer is accurate according to the context) and helpfulness (whether the response answers the question).

**[Requirement]** First provide analyses to all the answers, then assign each an integer between 1 and 10, where 1 means the answer is worst and 10 means the answer is perfect.

`{examples}`

`{context}`

Query:
`{query}`

Answers:
`{answers}`

---

**Aggregate Function Prompt**

**[Requirement]** Based on the provided current knowledge base, the input, output, and the score from the previous round, reconsider the following:

1. Which information from the knowledge base is necessary to solve the current problem and optimize the output, and which information is redundant.

2. Are there any errors in the information from the knowledge base?

After your consideration, reorganize the necessary information you plan to use, and remove any incorrect information. Directly output the consolidated result without additional instructions.

`{knowledge information}`

`{context}`

Answers:
`{answers}`

---

D.2    SURROGATE REWARD MODEL

**Reward Model Prompts**

**[Task]** Suppose that you are an expert in `{task_field}`, please rate an RLHF data pair consisting of a query, positive response and negative response.

**[Guideline]** Reference criteria:
1. The positive response should be coherent and correct as possible;

2. The negative response should be worse than the positive one in certain way, but not wander off the topic or diverge in too many aspects. For example, if the positive response is "The capital of France is Paris", a good negative response should be something like "The capital of France is London", but not "France is a country in Europe" (diverge too much in topic) or "Capital France London is" (diverge both in knowledge and language).

[**Requirement**] Please provide an integer score between 1 and 10 indicating the quality of the data pair if used in RLHF. The higher the score, the better the data pair. Please first analyze the positive response and the negative response, and then give the score in the format of "score/10".

{examples}

{context}

Query: {query}

Positive Response: {positive}
Negative Response: {negative}

### D.3 DETAILS OF MANUAL EVALUATION

For the evaluation of the difficulty of preference data pairs: We classified the difficulty of preference data pairs into four categories: very easy, easy, medium, and hard. The difficulty evaluation is mainly based on:

1. The difference between the preferred data and the dispreferred data in the preference pair. The smaller the difference, the higher the difficulty.
2. The difficulty of the question itself.

For the evaluation of the satisfaction level of the dataset: The evaluation is primarily based on the correctness of the preference data pair. For a preference data pair, if both the preferred data is correct and the dispreferred data is incorrect, it is marked as "Satisfied". If one of them is incorrect, it is marked as "Okay". Otherwise, it is marked as "Dissatisfied".

