# OpenReview forum: "Anyprefer: An Agentic Framework for Preference Data Synthesis"
_ICLR.cc/2025/Conference — ICLR 2025 Poster_

### Official Review · Reviewer_MDCN · 2024-10-19

**Soundness:** 3
**Presentation:** 4
**Contribution:** 3
**Rating:** 6
**Confidence:** 4

**Summary:**

The paper proposes Anyprefer, a novel framework for generating preference data for aligning AI models. Anyprefer accomplishes the preference data generation process as a cooperative two-player Markov Game between a target model and a judge model. The judge model uses external tools to get rewards and rank the responses generated by the target model. Additionally, a feedback mechanism is adopted to learn the prompts for both models. Experimental results show the effectiveness of Anyprefer comparison to existing baselines such as an untrained model and self-rewarding. The author also conducts a series of analyses to investigate different components of Anyprefer.

**Strengths:**

1. The paper is well-written.
2. The experiments are good and provide positive results on AnyPrefer.
3. The author claims to release a preference dataset that may help the community.
4. The framework is novel with the feedback learning mechanism and tool argumentation.

**Weaknesses:**

1. The author may provide at least a short explanation on how you update the prompt for GPT-4o (a black box model) with policy gradient? I suspect that not all audients are not famaliar with TextGrad. It is better to provide more experimental details in the appendix on how TextGrad is accomplished in different datasets.
2. The method requires an additional reward model for refining the prompt, what's the expected performance if directly using the additional reward model as the judge model?
3. The proposed method is called AnyPreference and the author conducts experiments on various scenarios that need preference data. The comparison and insights on evaluations of different scenarios can be discussed.
4. The author can consider more references regarding reward modeling, tool argumentation, and preference data construction, including but not limited to:
   Li L, Chai Y, Wang S, et al. Tool-augmented reward modeling[J]. arXiv preprint arXiv:2310.01045, 2023.
   Ye Z, Li X, Li Q, et al. Beyond Scalar Reward Model: Learning Generative Judge from Preference Data[J]. arXiv preprint arXiv:2410.03742, 2024.
   Mohammad Gheshlaghi Azar, Zhaohan Daniel Guo, Bilal Piot, Remi Munos, Mark Rowland,
   Michal Valko, and Daniele Calandriello. A general theoretical paradigm to understand learning from human preferences. In International Conference on Artificial Intelligence and Statistics,
   pp. 4447–4455. PMLR, 2024.
​  Junlong Li, Shichao Sun, Weizhe Yuan, Run-Ze Fan, Hai Zhao, and Pengfei Liu. Generative judge for evaluating alignment. arXiv preprint arXiv:2310.05470, 2023
  Wu T, Yuan W, Golovneva O, et al. Meta-rewarding language models: Self-improving alignment with llm-as-a-meta-judge[J]. arXiv preprint arXiv:2407.19594, 2024.
5. The method can be compared with more strong baselines in addition to an untrained model and self-rewarding, such as meta-rewarding.

**Questions:**

1. The method requires an additional reward model for refining the prompt, what's the expected performance if directly using the additional reward model as the judge model?
2. Are there any insights on the adoption of AnyPrefer in different scenarios?
3. If I understand correctly, the novelty of AnyPrefer lies in tool usage and prompt tunning from reward, these contribution looks more engineering-oriented. Is there any connection between the design of these two modules?

---

> ### Author Response · Authors · 2024-11-22
> **Response to Reviewer MDCN (1/2)**
>
> Thank you for your constructive comments and suggestions. We have revised our paper according to your comments. We respond to your questions below and would appreciate it if you could let us know if our response addresses your concerns.
> ****
> > **Q1**: The author may provide at least a short explanation on how you update the prompt for GPT-4o (a black box model) with policy gradient? I suspect that not all audients are not famaliar with TextGrad. It is better to provide more experimental details in the appendix on how TextGrad is accomplished in different datasets.
>
> **A1**: Thank you for pointing this out. Our approach to updating the prompts for the judge model and input prompt is similar to the method used in TextGrad [1], where the entire process of prompt iteration and updating is analogized to gradient descent. Specifically, when the reward model assigns a score below the threshold to the currently generated preference data, it provides feedback to update the input prompt for the target model and the tool information for the judge model. These updates are directly applied as textual modifications. To clarify this further, we have added the corresponding explanation in Section 2 of the revised paper.
> ****
> > **Q2**: The method requires an additional reward model for refining the prompt, what's the expected performance if directly using the additional reward model as the judge model?
>
> **A2**: Thank you for the reviewer’s question. In our experiments, both the reward model and the judge model are implemented using GPT-4o. The only difference between them lies in their functionalities and prompts, as explained in detail in the experimental setup section (Section 3). Specifically, the reward model is used to evaluate whether the preference data pairs generated and selected by the judge model and target model meet the given criteria. On the other hand, the judge model is responsible for ranking the responses sampled from the target model to construct the preference data pairs. We would like to highlight that, since each model uses a distinct prompt, they are treated as separate models.
> ****
> > **Q3**: Are there any insights on the adoption of AnyPrefer in different scenarios?
>
> **A3**: Thank you for the reviewer’s question. First, in the era of large models, manually annotating data is extremely time-consuming and labor-intensive. The automated synthesis of high-quality data holds significant importance for the development of large models today. At the same time, preference learning is crucial not only for LLMs but also for LVLMs. Currently, the most scarcest resource in preference learning is high-quality preference data. This work aims to automatically construct high-quality preference data, which is of significant importance to the development of this field. It can be observed that the data constructed by Anyprefer is effective across different domains, especially in the fields of Medical Image Analysis and Visuo-Motor Control. This highlights the importance of high-quality preference data for further improving model performance in specialized domains. Additionally, the ablation studies and manual evaluations demonstrate that incorporating tools significantly enhances the quality of the generated preference data. This suggests that integrating more precise tool information to improve the scoring accuracy of the judge model can greatly contribute to generating higher-quality preference data.
> ****
> > **Q4**: The author can consider more references regarding reward modeling, tool argumentation, and preference data construction, including but not limited to: Li L, Chai Y, Wang S, et al. Tool-augmented reward modeling[J]. arXiv preprint arXiv:2310.01045, 2023. Ye Z, Li X, Li Q, et al. Beyond Scalar Reward Model: Learning Generative Judge from Preference Data[J]. arXiv preprint arXiv:2410.03742, 2024. Mohammad Gheshlaghi Azar, Zhaohan Daniel Guo, Bilal Piot, Remi Munos, Mark Rowland, Michal Valko, and Daniele Calandriello. A general theoretical paradigm to understand learning from human preferences. In International Conference on Artificial Intelligence and Statistics, pp. 4447–4455. PMLR, 2024. Junlong Li, Shichao Sun, Weizhe Yuan, Run-Ze Fan, Hai Zhao, and Pengfei Liu. Generative judge for evaluating alignment. arXiv preprint arXiv:2310.05470, 2023 Wu T, Yuan W, Golovneva O, et al. Meta-rewarding language models: Self-improving alignment with llm-as-a-meta-judge[J]. arXiv preprint arXiv:2407.19594, 2024.
>
> **A4**: We have carefully reviewed these papers, and they are indeed highly relevant to our work. Thank you for your suggestion. We have incorporated discussions of these articles in the revised version of the manuscript.
> ****

---

> ### Author Response · Authors · 2024-11-22
> **Response to Reviewer MDCN (2/2)**
>
> > **Q5**: The method can be compared with more strong baselines in addition to an untrained model and self-rewarding, such as meta-rewarding.
>
> **A5**: Thank you for your suggestion. Due to the inapplicability of the prompt design in Meta-rewarding for the Visuo-Motor Control domain, we have added Meta-rewarding as a baseline in the new version of the manuscript for the other three domains: Natural Language Generation, Natural Vision-Language Understanding, and Medical Image Analysis. Additionally, we have discussed this method in the experiment and related work section. We report the results in the following three Tables (Table R12 - R14). According to the results, we observe that Anyprefer outperforms the meta-rewarding method in all tasks. Though Meta-rewarding improved the rewarding bias issue of self-rewarding, it is inferior than Anyprefer incorporating external tools and learning from feedback mechanism to mitigate this issue.
>
> **Table R12**: Results on natural language generation.
> | **Method**               | **GSM8K** | **ARC-Easy** | **ARC-Challenge** | **Alpaca Eval 2.0** |
> |--------------------------|-----------|--------------|-------------------|---------------------|
> | LLaMA-2                 | 22.44     | 74.33        | 57.68            | 5.20 / 4.57         |
> | + Self Rewarding        | 23.20     | 74.45        | 56.31            | 3.28 / 3.12                  |
> | + Meta Rewarding | 25.47     | 76.22        | 59.47            | -                   |
> | + Anyprefer              | **38.14** | **80.26**    | **64.68**        | **19.25 / 15.14**   |
>
> **Table R13**: Results on natural vision-language understanding.
>
> | **Method**           | **MME^P** | **MME^C** | **LLAVA^W** | **MMB** | **MMVet** | **SQA^I** | **VisWiz** | **GQA** | **POPE** |
> |-----------------------|-----------|-----------|-------------|---------|-----------|-----------|------------|---------|----------|
> | LLaVA-1.5-7B         | **1510.7**| 348.2     | 63.4        | 64.3    | 30.5      | 66.8      | 50.0       | 62.0    | 85.90    |
> | + Self Rewarding      | 1505.6    | 362.5     | 61.2        | 64.5    | 31.4      | 69.6      | 53.9       | 61.7    | 86.88    |
> | + Meta Rewarding      | 1498.3    | 357.4     | 64.0        | 64.2    | 31.3      | 69.1      | 53.5       | 62.0    | 86.70    |
> | + Anyprefer           | 1510.1    | **362.9** | **69.2**    | **65.1**| **33.0**  | **70.9**  | 54.0       | 62.2    | **86.98**|
>
> **Table R14**: Results on medical image analysis.
> | **Method**            | **VQA-RAD (Closed)** | **VQA-RAD (Open)** | **SLAKE (Closed)** | **SLAKE (Open)** | **BLEU-1** | **BLEU-2** | **BLEU-3** | **BLEU-4** | **ROUGE-L** | **METEOR** |
> |------------------------|---------------------|--------------------|--------------------|------------------|------------|------------|------------|------------|-------------|------------|
> | LLaVA-Med             | 63.57              | 32.09             | 61.30             | 44.26           | 10.31      | 0.66       | 0.07       | 0.01       | 10.32       | 10.95      |
> | + Self Rewarding      | 64.17              | 32.29             | 61.30             | 42.63           | 9.71       | 0.97       | 0.10       | 0.01       | 10.38       | 10.52      |
> | + Meta Rewarding       | 67.42              | 33.05             | 65.10             | 45.12           | 12.48      | 1.23       | 0.24       | 0.03       | 12.56       | 13.21      |
> | + Anyprefer            | **72.06**          | **36.10**         | **70.39**         | **49.04**       | **16.85**  | **5.57**   | **2.07**   | **0.56**   | **23.69**   | **29.66**  |
> ****
> > **Q6**: If I understand correctly, the novelty of AnyPrefer lies in tool usage and prompt tunning from reward, these contribution looks more engineering-oriented. Is there any connection between the design of these two modules?
>
> **A6**: Thank you for the reviewer’s question. In Anyprefer, the tool usage module and prompt tuning module work synchronously. The tool usage can improve the accuracy of information to assist the rewarding, while this process can be further improved if we could selectively use the tools and effectively aggreate the information. Here, our prompt optimization comes to contribute to this. Specifically, as shown in Algorithm 1, when the constructed preference data does not meet the requirements (i.e., falls below the threshold $\tau$), Anyprefer updates and optimizes the prompts $\mathbf{p}_t$ and $\mathbf{p}_j$. The prompt $\mathbf{p}_t$ includes information extracted by the tools, which is updated using the prompt provided in Appendix C.2. Through feedback updates, the tool-information integration mechanism becomes more accurate. This results in higher-quality preference data generated by Anyprefer.
> ****
>
> **Reference**:
>
> [1] Yuksekgonul, M., Bianchi, F., Boen, J., Liu, S., Huang, Z., Guestrin, C., & Zou, J. “TextGrad: Automatic ‘Differentiation’ via Text.” arXiv preprint arXiv:2406.07496 (2024).

---

> > ### Comment · Reviewer_MDCN · 2024-11-25
> >
> > The author's response is good. The paper still has some issues, such as the lack of connection between the two novelties and the unclear description of the two-player game as the paper's starting point. However, considering the nice writing and the good baseline experiments added during the rebuttal, I am open to hearing the opinions of other reviewers. At this stage, I plan to maintain my score because I have already given a positive rating.

---

> > > ### Author Response · Authors · 2024-11-26
> > > **Additional Response to Reviewer MDCN**
> > >
> > > Dear Reviewer MDCN,
> > >
> > > Thank you very much for your time and efforts in reviewing our response and engaging in the discussion! We greatly appreciate your recognition of the nice writing and the good baseline experiments added during the rebuttal. We are happy to further clarify your remaining concerns below:
> > >
> > > > lack of connection between the two novelties
> > >
> > > We would like to further clarify the connection between these two novelties. The interplay between tool usage and prompt optimization (prompt tuning) serves as the key link connecting the two. Specifically, tool usage provides the necessary information for the judge model to make quality judgments, and these judgments are used to optimize the prompts. The prompt optimization process not only updates the prompts for the target model’s input but also adjusts the information extracted by the tools (as not all tool-generated information is accurate or important). This iterative process allows prompt optimization and tool usage to mutually enhance each other, enabling the synthesis of higher-quality preference data.
> > >
> > > > unclear description of the two-player game as the paper's starting point
> > >
> > > Our goal in formulating this problem as a two-player cooperative game is to provide a prototype framework for understanding the preference data synthesis process. The formulation is motivated by the fact that the universal goal: maximizing the quality of the preference data, cannot be simply achieved by solely improving the target model (generative data) and judge model (generate preference). This goal requires collaboration between the target model and the judge model in generating the response and scoring them. Together, they produce a higher-quality preference dataset than either model could achieve independently. Furthermore, the iterative feedback loop from the reward model ensures mutual benefit and stability in their collaboration. We have revised the introduction to further clarify it.
> > >
> > > Thank you once again for taking the time to discuss with us. We would greatly appreciate it if you could let us know of any further concerns.

---

### Official Review · Reviewer_zGJr · 2024-11-01

**Soundness:** 3
**Presentation:** 3
**Contribution:** 3
**Rating:** 6
**Confidence:** 3

**Summary:**

This paper proposed Anyprefer, an automatic method to synthesize high-quality preference data to improve model alignment with human values. The preference data generation is set as a cooperative two-player Markov game between a target model and a judge model. Using external tools and feedback, the proposed Anyprefer claim to show substantial performance improvement across four applications: natural language generation, vision-language understanding, medical image analysis, and visuo-motor control.

**Strengths:**

-  This paper solves an important problem in synthetic data generation for model alignment. The proposed framework can automates preference data synthesis, which can reduce the reliance on human annotations.
- The proposed model uses external tools in the reward phase, which may be able to make the data more diverse, reduce bias, and improve data quality in preference synthesis.
- The paper conducted extensive experiments and ablation studies across multiple applications.

**Weaknesses:**

- The setting, a two-player cooperative Markov Game for preference data synthesis, is only vaguely justified. The paper could be better if adding some explanations about  how this approach explicitly improves alignment over simpler, existing methods.
- External tools are incorporated to reduce bias. It would be good to add additional analysis or metrics to show that these tools effectively mitigate model bias or improve reliability beyond existing methods.
- The proposed model’s effectiveness is evaluated on a narrow set of datasets within each application domain, which may not be able to show its overall generalizability. It would be good to perform on datasets or domains outside the four tested applications as well.
- While the  feedback mechanisms are described,  the impact of these mechanisms on the process is not clearly analyzed. It would be good to add ablation study to study their specific contribution.

**Questions:**

Please see the comments above.

---

> ### Author Response · Authors · 2024-11-22
> **Response to Reviewer zGJr (1/5)**
>
> Thank you for your constructive comments and suggestions. We have revised our paper according to your comments. We respond to your questions below and would appreciate it if you could let us know if our response addresses your concerns.
> ****
> > **Q1**: The setting, a two-player cooperative Markov Game for preference data synthesis, is only vaguely justified. The paper could be better if adding some explanations about how this approach explicitly improves alignment over simpler, existing methods.
>
> **A1**: Thank you for your question. We framed the generation process as a “two-player game” primarily to provide a more intuitive explanation of the interaction and collaboration involved in constructing and optimizing preference data. In this framework, the judge model and the target model are viewed as two players, each playing a distinct role: the judge model evaluates the generated responses and constructs preference data, while the target model improves based on this preference data. Together, they collaborate to generate preference data, which is assessed by the reward model to determine the data quality. This reward is then used to enhance both models and their collaboration, ultimately leading to better preference data generation. By framing the process as a "two-player game", we emphasize the importance of the cooperative relationship between the judge model and the target model.
>
> To better demonstrate the benefits of the “two-player game” framework and the importance of collaboration between the judge model and the target model, we conducted an ablation study with the following ablation models:
> - (1) Optimizing only the input prompt for the target model using the reward.
> - (2) Optimizing only the prompt for the judge model using the reward.
> - (3) Independently optimizing the target model’s prompt and the judge model’s information, then combining them. In this ablation model, by utilizing an independent optimization mechanism, we partially eliminate the collaboration between the target model and the judge model.
>
> The results of four domains are reported in Table R1, R2, R3 and R4 respectively. The results suggest that jointly updating the judge model and the reward model leads to greater performance improvements than optimizing them individually.

---

> ### Author Response · Authors · 2024-11-22
> **Response to Reviewer zGJr (2/5)**
>
> **Table R1**: Ablation study on natural language generation.
> | **Method**                  | **GSM8K** | **ARC-Easy** | **ARC-Challenge** | **Alpaca Eval 2.0**  |
> |-----------------------------|-----------|--------------|-------------------|----------------------|
> | (1) Optimize Target Model Only | 28.12     | 76.02        | 59.12            | 14.58 / 12.34        |
> | (2) Optimize Judge Model Only  | 29.25     | 77.56        | 60.45            | 15.32 / 13.12        |
> | (3) Independently Optimize Both Models | 32.18     | 78.90        | 61.98            | 17.04 / 14.02        |
> | **Anyprefer (Ours)** | **38.14** | **80.26**    | **64.68**        | **19.25 / 15.14**    |
>
> **Table R2**: Ablation study on natural vision-language understanding.
> | **Method**                  | **MME^P** | **MME^C** | **LLAVA^W** | **MMB** | **MMVet** | **SQA^I** | **VisWiz** | **GQA** | **POPE** |
> |-----------------------------|-----------|-----------|-------------|---------|-----------|-----------|------------|---------|----------|
> | (1) Optimize Target Model Only | 1480.2    | 350.2     | 65.2        | 63.9    | 31.0      | 67.2      | 52.0       | 61.5    | 86.10    |
> | (2) Optimize Judge Model Only  | 1485.6    | 353.1     | 66.1        | 64.1    | 31.5      | 68.0      | 53.2       | 61.8    | 86.45    |
> | (3) Independently Optimize Both Models | 1495.8    | 359.0     | 67.8        | 64.7    | 32.5      | 69.2      | 53.5       | 62.0    | 86.70    |
> | **Anyprefer (Ours)** | **1510.1**| **362.9** | **69.2**    | **65.1**| **33.0**  | **70.9**  | **54.0**   | **62.2**| **86.98**|
>
> **Table R3**: Ablation study on medical image analysis.
> | **Method**                | **VQA-RAD (Closed)** | **VQA-RAD (Open)** | **SLAKE (Closed)** | **SLAKE (Open)** | **BLEU-1** | **BLEU-2** | **BLEU-3** | **BLEU-4** | **ROUGE-L** | **METEOR** |
> |---------------------------|---------------------|--------------------|--------------------|------------------|------------|------------|------------|------------|-------------|------------|
> | (1) Optimize Target Model Only |  66.20   |   33.53   |     65.36    |  45.79  |   12.89   |   3.16    |     1.20  |  0.28 | 14.87  | 17.99
> | (2) Optimize Judge Model Only  | 68.71 |  34.39   |   67.48   |   46.60  |   13.37    |    3.79  |   1.42   |   0.36  |   17.89  | 19.60
> | (3) Independently Optimize Both Models | 70.21  | 34.89    |    68.37     |  47.65  |   14.57  |   4.18   |   1.68    |  0.43  |  20.66  | 23.74
> | + Anyprefer (Ours)              | **72.06**          | **36.10**         | **70.39**         | **49.04**       | **16.85**  | **5.57**   | **2.07**   | **0.56**   | **23.69**   | **29.66**  |
>
> **Table R4**: Ablation study on visuo-motor control.
> | **Method**                     | **Grasp Spoon** | **Success (Spoon)** | **Grasp Carrot** | **Success (Carrot)** | **Grasp Cube** | **Success (Cube)** | **Grasp Eggplant** | **Success (Eggplant)** |
> |--------------------------------|-----------------|---------------------|------------------|----------------------|----------------|--------------------|--------------------|------------------------|
> | (1) Optimize Target Model Only |      0.53       |        0.34         |      0.46        |         0.28         |      0.57      |        0.22        |        0.60        |          0.42          |
> | (2) Optimize Judge Model Only  |      0.53       |        0.36         |      0.49        |         0.30         |      0.58      |        0.24        |        0.62        |          0.44          |
> | (3) Independently Optimize Both Models | 0.55  |  0.38   |   0.52   |   0.33  |   0.59   |   0.26   |   0.62   |   0.47   |
> | + Anyprefer (Ours)              | **0.56**       |     **0.40**       |     **0.54**     |       **0.44**       |    **0.61**    |      **0.28**      |      **0.66**      |        **0.50**        |
> ****
> > **Q2**: External tools are incorporated to reduce bias. It would be good to add additional analysis or metrics to show that these tools effectively mitigate model bias or improve reliability beyond existing methods.
>
> **A2**: We conducted an ablation study in the domain of Natural Vision-Language Understanding to analyze the contribution of different tools to this domain. Specifically, we have added detailed results in Appendix B and referenced them in the main text under the experimental section. We reported the results in Table R5 and R6. According to the results, for both tasks, tools capable of extracting entity-level information (e.g., disease detection) from images outperform description-based tools (e.g., image captioning tools). Furthermore, combining multiple tools proves to be more effective than relying on a single tool, as it enables the utilization of richer and more diverse information during the generation of preference data. This underscores the importance of integrating the complementary capabilities of different tools.

---

> ### Author Response · Authors · 2024-11-22
> **Response to Reviewer zGJr (3/5)**
>
> **Table R5**: Effect of different tool combination on natural vision-language understanding.
> | **Method**                  | **MME^P** | **MME^C** | **LLAVA^W** | **MMB** | **MMVet** | **SQA^I** | **VisWiz** | **GQA** | **POPE** |
> |-----------------------------|-----------|-----------|-------------|---------|-----------|-----------|------------|---------|----------|
> | LLaVA-1.5-7B               | **1510.7**| 348.2     | 63.4        | 64.3    | 30.5      | 66.8      | 50.0       | 62.0    | 85.90    |
> | + Anyprefer (Florence-2-large)           | 1468.3    | 347.8     | 67.0        | 64.2    | 31.7      | 68.6      | 53.4       | 61.9    | 86.52    |
> | + Anyprefer (BLIP-2)                     | 1465.2    | 343.5     | 66.4        | 64.0    | 31.3      | 68.1      | 53.0       | 61.6    | 86.40    |
> | + Anyprefer (Grounded SAM)               | 1475.3    | 355.6     | 68.9        | 64.7    | 32.8      | 69.8      | **54.8**   | 62.0    | 86.75    |
> | + Anyprefer                | 1510.1    | **362.9** | **69.2**    | **65.1**| **33.0**  | **70.9**  | 54.0       | **62.2**| **86.98**|
>
> **Table R6**: Effect of different tool combination on medical image analysis.
> | **Method**                | **VQA-RAD (Closed)** | **VQA-RAD (Open)** | **SLAKE (Closed)** | **SLAKE (Open)** | **BLEU-1** | **BLEU-2** | **BLEU-3** | **BLEU-4** | **ROUGE-L** | **METEOR** |
> |---------------------------|---------------------|--------------------|--------------------|------------------|------------|------------|------------|------------|-------------|------------|
> | LLaVA-Med                | 63.57              | 32.09             | 61.30             | 44.26           | 10.31      | 0.66       | 0.07       | 0.01       | 10.32       | 10.95      |
> | Anyprefer (MiniGPT-Med) |  68.02  |  34.74  | 63.72      | 45.80  | 12.82     |  2.10   |  0.89   | 0.23 | 15.80 | 17.42
> | Anyprefer (MedVInT)  | 67.55 |  33.70 |   63.87  | 46.72   |  14.44   |  3.42    |  1.30   |  0.29  | 18.57   | 19.09
> | Anyprefer (CheXagent) | 69.06  | 35.48   |   64.50     |  46.61  |  13.10   |   3.99   |  1.46   |  0.35  |  20.30  | 22.17
> | + Anyprefer              | **72.06**          | **36.10**         | **70.39**         | **49.04**       | **16.85**  | **5.57**   | **2.07**   | **0.56**   | **23.69**   | **29.66**  |
> ****
> > **Q3**: The proposed model’s effectiveness is evaluated on a narrow set of datasets within each application domain, which may not be able to show its overall generalizability. It would be good to perform on datasets or domains outside the four tested applications as well.
>
> **A3**: Thank you for the reviewer’s suggestion. To evaluate the model’s performance in additional application domains, we referred to [1]. Using the pre-trained model on LLaMA-2 provided in [1] as the target model within Anyprefer, we randomly sampled 2,000 examples from the MIMIC-IV-ECG dataset, collected 2,000 preference data pairs, and applied Anyprefer to these data. For comparison, we also evaluated the performance of self-rewarding on the same dataset. Subsequently, we assessed their performance on the electrocardiogram (ECG) report generation task. The results are summarized in Table R7.
>
> According to the results, we can observe that Anyprefer also performs well on the electrocardiogram (ECG) report generation task, achieving slight improvements over the baseline (LLaMA-2) across various metrics. This further validates the applicability and effectiveness of Anyprefer in different application domains.

---

> ### Author Response · Authors · 2024-11-22
> **Response to Reviewer zGJr (4/5)**
>
> **Table R7**: Performance on electrocardiogram (ECG) report generation.
> | **Model**          | **BLEU-1** | **BLEU-2** | **BLEU-3** | **BLEU-4** | **METEOR** | **ROUGE-L** | **ROUGE-1** | **CIDEr-D** |
> |---------------------|------------|------------|------------|------------|------------|-------------|-------------|-------------|
> | LLaMA-2            | 0.706      | 0.662      | 0.622      | 0.581      | 0.775      | 0.745       | 0.765   | 5.55        |
> | + Self-Rewarding      | 0.711      | 0.664      | 0.620      | 0.582      | 0.767      | 0.744       | **0.768**   | 5.56        |
> | + Anyprefer         | **0.726**  | **0.674**  | **0.628**  | **0.585**  | **0.781**  | **0.753**   | 0.763       | **5.65**    |
> ****
> > **Q4**: While the feedback mechanisms are described, the impact of these mechanisms on the process is not clearly analyzed. It would be good to add ablation study to study their specific contribution.
>
> **A4**: Thank you very much for the reviewer’s valuable suggestion. We would like to clarify that we have already analyzed the specific contribution of the feedback mechanism through an ablation study in the main text, with the relevant results presented in Section 3.3, Table 2. In this study, we observe that removing the feedback mechanism leads to significant performance drops across four domains (Natural Language Generation, Natural Vision-Language Understanding, Medical Image Analysis, and Visuo-Motor Control). This validates the importance of the feedback mechanism for Anyprefer.
>
> In addition, to better understand the impact of feedback mechanism, we conducted an ablation study with the following ablation models:
> - (1) Optimizing only the input prompt for the target model using the reward.
> - (2) Optimizing only the prompt for the judge model using the reward.
>
> The results of four domains are reported in Table R8, R9, R10 and R11, respectively. The results suggest that jointly updating the judge model and the reward model leads to greater performance improvements than optimizing them individually.

---

> ### Author Response · Authors · 2024-11-22
> **Response to Reviewer zGJr (5/5)**
>
> **Table R8**: Ablation study on natural language generation.
> | **Method**                  | **GSM8K** | **ARC-Easy** | **ARC-Challenge** | **Alpaca Eval 2.0**  |
> |-----------------------------|-----------|--------------|-------------------|----------------------|
> | (1) Optimize Target Model Only | 28.12     | 76.02        | 59.12            | 14.58 / 12.34        |
> | (2) Optimize Judge Model Only  | 29.25     | 77.56        | 60.45            | 15.32 / 13.12        |
> | **Anyprefer (Ours)** | **38.14** | **80.26**    | **64.68**        | **19.25 / 15.14**    |
>
> **Table R9**: Ablation study on natural vision-language understanding.
> | **Method**                  | **MME^P** | **MME^C** | **LLAVA^W** | **MMB** | **MMVet** | **SQA^I** | **VisWiz** | **GQA** | **POPE** |
> |-----------------------------|-----------|-----------|-------------|---------|-----------|-----------|------------|---------|----------|
> | (1) Optimize Target Model Only | 1480.2    | 350.2     | 65.2        | 63.9    | 31.0      | 67.2      | 52.0       | 61.5    | 86.10    |
> | (2) Optimize Judge Model Only  | 1485.6    | 353.1     | 66.1        | 64.1    | 31.5      | 68.0      | 53.2       | 61.8    | 86.45    |
> | **Anyprefer (Ours)** | **1510.1**| **362.9** | **69.2**    | **65.1**| **33.0**  | **70.9**  | **54.0**   | **62.2**| **86.98**|
>
>
> **Table R10**: Ablation study on medical image analysis.
> | **Method**                | **VQA-RAD (Closed)** | **VQA-RAD (Open)** | **SLAKE (Closed)** | **SLAKE (Open)** | **BLEU-1** | **BLEU-2** | **BLEU-3** | **BLEU-4** | **ROUGE-L** | **METEOR** |
> |---------------------------|---------------------|--------------------|--------------------|------------------|------------|------------|------------|------------|-------------|------------|
> | (1) Optimize Target Model Only |  66.20   |   33.53   |     65.36    |  45.79  |   12.89   |   3.16    |     1.20  |  0.28 | 14.87  | 17.99
> | (2) Optimize Judge Model Only  | 68.71 |  34.39   |   67.48   |   46.60  |   13.37    |    3.79  |   1.42   |   0.36  |   17.89  | 19.60
> | **Anyprefer (Ours)**              | **72.06**          | **36.10**         | **70.39**         | **49.04**       | **16.85**  | **5.57**   | **2.07**   | **0.56**   | **23.69**   | **29.66**  |
>
> **Table R11**: Ablation study on visuo-motor control.
> | **Method**                     | **Grasp Spoon** | **Success (Spoon)** | **Grasp Carrot** | **Success (Carrot)** | **Grasp Cube** | **Success (Cube)** | **Grasp Eggplant** | **Success (Eggplant)** |
> |--------------------------------|-----------------|---------------------|------------------|----------------------|----------------|--------------------|--------------------|------------------------|
> | (1) Optimize Target Model Only |      0.53       |        0.34         |      0.46        |         0.28         |      0.57      |        0.22        |        0.60        |          0.42          |
> | (2) Optimize Judge Model Only  |      0.53       |        0.36         |      0.49        |         0.30         |      0.58      |        0.24        |        0.62        |          0.44          |
> | **Anyprefer (Ours)**              | **0.56**       |     **0.40**       |     **0.54**     |       **0.44**       |    **0.61**    |      **0.28**      |      **0.66**      |        **0.50**        |
> ****
> **Reference**:
>
> [1] Wan, Z., Liu, C., Wang, X., Tao, C., Shen, H., Peng, Z., Fu, J., Arcucci, R., Yao, H., & Zhang, M. “MEIT: Multi-Modal Electrocardiogram Instruction Tuning on Large Language Models for Report Generation.” arXiv preprint arXiv:2403.04945 (2024).
> ****

---

### Official Review · Reviewer_xwba · 2024-11-03

**Soundness:** 3
**Presentation:** 1
**Contribution:** 3
**Rating:** 8
**Confidence:** 3

**Summary:**

In this paper, the authors consider the problem of how to generate a preference data dataset synthetically, as current methods for generating high-quality preference-data currently require significant data annotation. As a solution, the authors propose Anyprefer a framework that frames the preference dataset production as a two-player game between the target and judge models, with a third method acting as a reward proxy for what the performance increase would be if the target (base) model was optimized through preference learning. Anyprefer functions by the target model producing a set of candidate responses to some prompt p_t, which are then ranked by a judge model that additionally gets inputs from expert tools that correspond to the kind of dataset that is being optimized for (vision, medical, natural language generation, or control in their case). They then use a reward model that determines if the preference pair, constructed from the best and worst of the ranking, was sufficiently good to be included. The authors then demonstrate the benefits of such generated pairs on a set of downstream tasks in comparison to the original target models, and both a self-rewarding scheme versus their proposed method that are both used to generate pairs and which are subsequently  used for fine-tuning.

**Strengths:**

1. The authors consider quite a broad set of target models and target domains to demonstrate their method on, showing that it's efficacious in different data scenarios and regimes.

2. Anyprefer cleverly integrates other tools outside of the target and judge model, and because of the nature of the judge model being an incredibly powerful VLM like GPT-4o, this strategy seems like an extendable and scalable as more specialized tools become available.

3. The improvements on downstream target models, in comparison to baselines, is clear. This technique effectively is able to take advantage of GPT-4o to improve the capabilities of weaker models.

**Weaknesses:**

1. It is unclear in the paper what the authors gain in perspective or methodology from framing their generation as a "two-player game". Given that this technical detail was referenced several times, I was expecting that it would be involved in some sort of modeling decision or theoretical result. Another point that I believe is important to explain well, and lacked clarity, was how one can update the policy of the target model p_t, by optimizing the input prompt p_t. This mixes together a few notions from language modeling and RL that I believe should be explained better.

2. The authors should include their benchmarking on IMG/MED datasets with the synthetic dataset VLFeedback in the main text. If I understand correctly, it similarly does not require human effort, is larger than Anyprefer-V1, and could be compared against to demonstrate the claim made by Figure 6 that the diversity of Anyprefer is its strength over existing techniques. Additionally, I was not convinced by the t-SNE plots that Anyprefer seemed to cover all datasets, it is just in the distribution diagram spread that they appear either more sparse (Anyprefer) or more dense (VLFeedback). A quantitative analysis here consistent with what is done in the literature seems necessary.

3. The authors claim in Lines 70 and 95 that the methods used as 'tools' for providing auxiliary information to the judge models are 'selected based on the input prompt' or 'Anyprefer strategically selects tools'. From what I understand in section 3.1, either the tools are selected manually corresponding to the task or it is not explained how 'Anyprefer' selects tools. This makes me believe that how the tools are chosen is ambiguous.

**Questions:**

1. In line 47, its said that "…this approach may fail to capture the inherence preferences of the target model being fine-tuned, rendering the data-generation less useful". I found this phrasing quite confusing, as if your target model is imperfect, why would it matter that you aren't aligned with it. From my perspective the whole benefit of the proposed system is effectively a distillation of GPT-4o preferences into the target model.

2. It would've been interesting to see ablations regardless the dataset generation process. For example, if the Judge model produces quality rankings, then doesn't it seem suboptimal to use only the best and worst generation? Couldn't one use many different pairs (preserving ranking order for + or -) from the ranking? Also it seems that 5 was chosen as the number of generations, but it's unclear the role that this plays in the final quality.

3. Once the preference datasets are generated using any of the proposed techniques, what was the process for actually improving the target models? We are given the results but without any experimental setup for that tuning. I'm leaning on making this a weakness, as I think it's important to be included, but it could be that this should be a common knowledge process.

4. Why is it the case that both the prompt of the target model AND of the judge model need to be optimized in this process? Is it not sufficient to optimize the prompt of the target model through the preferences of the judge? It would be interesting to see that this is insufficient.

5. Where does so much improvement come from in the medical VQA task? It seems like 31.05% is a massive gain.

6. In section 3.5, how were these samples manually selected?

7. Was there an analysis done on how including different numbers of tools was helpful for different datasets, or which tools mattered the most?

---

> ### Author Response · Authors · 2024-11-22
> **Response to Reviewer xwba (1/8)**
>
> Thank you for reviewing our paper and for your valuable feedback. Below, we address your concerns point by point and we’ve revised our paper according to your suggestions. We would appreciate it if you could let us know whether your concerns are addressed by our response.
> ****
> > **Q1**: It is unclear in the paper what the authors gain in perspective or methodology from framing their generation as a "two-player game". Given that this technical detail was referenced several times, I was expecting that it would be involved in some sort of modeling decision or theoretical result.
>
>
> **A1**: Thank you for the reviewer’s question. We framed the generation process as a “two-player game” primarily to provide a more intuitive explanation of the interaction and collaboration involved in constructing and optimizing preference data. In this framework, the judge model and the target model are viewed as two players, each playing a distinct role: the judge model evaluates the generated responses and constructs preference data, while the target model improves based on this preference data. Together, they collaborate to generate preference data, which is assessed by the reward model to determine the data quality. This reward is then used to enhance both models and their collaboration, ultimately leading to better preference data generation. By framing the process as a "two-player game", we emphasize the importance of the cooperative relationship between the judge model and the target model.
>
> To better demonstrate the benefits of the “two-player game” framework and the importance of collaboration between the judge model and the target model, we conducted an ablation study with the following ablation models:
> - (1) Optimizing only the input prompt for the target model using the reward.
> - (2) Optimizing only the prompt for the judge model using the reward.
> - (3) Independently optimizing the target model’s prompt and the judge model’s information, then combining them. In this ablation model, by utilizing an independent optimization mechanism, we partially eliminate the collaboration between the target model and the judge model.
>
> We report the results on the four domains mentioned in the text in Table R1, Table R2, Table R3 and Table R4, respectively. The results indicate that individually optimizing the judge model and the reward model yields less performance improvement compared to jointly updating them, indicating the importance of collaboration between the judge model and the reward model.

---

> > ### Comment · Reviewer_xwba · 2024-11-25
> > **Response to (1/8)**
> >
> > I understand the motivation for framing the optimization process as a collaboration effort between the two models. I suppose my concern I raised was because 'two-player game' invokes relations to game-theoretic ideas that I don't see as present in the paper. I think that this distracts from the method itself rather than providing useful intuition.

---

> ### Author Response · Authors · 2024-11-22
> **Response to Reviewer xwba (2/8)**
>
> **Table R1**: Ablation study on natural language generation.
> | **Method**                  | **GSM8K** | **ARC-Easy** | **ARC-Challenge** | **Alpaca Eval 2.0**  |
> |-----------------------------|-----------|--------------|-------------------|----------------------|
> | (1) Optimize Target Model Only | 28.12     | 76.02        | 59.12            | 14.58 / 12.34        |
> | (2) Optimize Judge Model Only  | 29.25     | 77.56        | 60.45            | 15.32 / 13.12        |
> | (3) Independently Optimize Both Models | 32.18     | 78.90        | 61.98            | 17.04 / 14.02        |
> | **Anyprefer (Ours)** | **38.14** | **80.26**    | **64.68**        | **19.25 / 15.14**    |
>
> **Table R2**: Ablation study on natural vision-language understanding.
> | **Method**                  | **MME^P** | **MME^C** | **LLAVA^W** | **MMB** | **MMVet** | **SQA^I** | **VisWiz** | **GQA** | **POPE** |
> |-----------------------------|-----------|-----------|-------------|---------|-----------|-----------|------------|---------|----------|
> | (1) Optimize Target Model Only | 1480.2    | 350.2     | 65.2        | 63.9    | 31.0      | 67.2      | 52.0       | 61.5    | 86.10    |
> | (2) Optimize Judge Model Only  | 1485.6    | 353.1     | 66.1        | 64.1    | 31.5      | 68.0      | 53.2       | 61.8    | 86.45    |
> | (3) Independently Optimize Both Models | 1495.8    | 359.0     | 67.8        | 64.7    | 32.5      | 69.2      | 53.5       | 62.0    | 86.70    |
> | **Anyprefer (Ours)** | **1510.1**| **362.9** | **69.2**    | **65.1**| **33.0**  | **70.9**  | **54.0**   | **62.2**| **86.98**|
>
> **Table R3**: Ablation study on medical image analysis.
> | **Method**                | **VQA-RAD (Closed)** | **VQA-RAD (Open)** | **SLAKE (Closed)** | **SLAKE (Open)** | **BLEU-1** | **BLEU-2** | **BLEU-3** | **BLEU-4** | **ROUGE-L** | **METEOR** |
> |---------------------------|---------------------|--------------------|--------------------|------------------|------------|------------|------------|------------|-------------|------------|
> | (1) Optimize Target Model Only |  66.20   |   33.53   |     65.36    |  45.79  |   12.89   |   3.16    |     1.20  |  0.28 | 14.87  | 17.99
> | (2) Optimize Judge Model Only  | 68.71 |  34.39   |   67.48   |   46.60  |   13.37    |    3.79  |   1.42   |   0.36  |   17.89  | 19.60
> | (3) Independently Optimize Both Models | 70.21  | 34.89    |    68.37     |  47.65  |   14.57  |   4.18   |   1.68    |  0.43  |  20.66  | 23.74
> | + Anyprefer (Ours)              | **72.06**          | **36.10**         | **70.39**         | **49.04**       | **16.85**  | **5.57**   | **2.07**   | **0.56**   | **23.69**   | **29.66**  |
>
> **Table R4**: Ablation study on visuo-motor control.
> | **Method**                     | **Grasp Spoon** | **Success (Spoon)** | **Grasp Carrot** | **Success (Carrot)** | **Grasp Cube** | **Success (Cube)** | **Grasp Eggplant** | **Success (Eggplant)** |
> |--------------------------------|-----------------|---------------------|------------------|----------------------|----------------|--------------------|--------------------|------------------------|
> | (1) Optimize Target Model Only |      0.53       |        0.34         |      0.46        |         0.28         |      0.57      |        0.22        |        0.60        |          0.42          |
> | (2) Optimize Judge Model Only  |      0.53       |        0.36         |      0.49        |         0.30         |      0.58      |        0.24        |        0.62        |          0.44          |
> | (3) Independently Optimize Both Models | 0.55  |  0.38   |   0.52   |   0.33  |   0.59   |   0.26   |   0.62   |   0.47   |
> | + Anyprefer (Ours)              | **0.56**       |     **0.40**       |     **0.54**     |       **0.44**       |    **0.61**    |      **0.28**      |      **0.66**      |        **0.50**        |
> ****

---

> ### Author Response · Authors · 2024-11-22
> **Response to Reviewer xwba (3/8)**
>
> > **Q2**: Another point that I believe is important to explain well, and lacked clarity, was how one can update the policy of the target model p_t, by optimizing the input prompt p_t. This mixes together a few notions from language modeling and RL that I believe should be explained better.
>
> **A2**: Thank you very much for the reviewer’s question. First, we would like to clarify that the optimization process of our prompts refinement by feedback is similar to updating the parameters via gradient descent, though the gradient is not actually passing through the textual prompt. This concept is drawn from TextGrad[1], and we will revise related descriptions to make it more clear. The detailed optimization process is explained as follows:
>
> * In Anyprefer, we optimize both the input prompt of the target model and the tool information utilized by the judge model through a feedback mechanism. This process indirectly updates the target model's policy to generate candidate responses while enhancing the judge model's ability to integrate tool information effectively.
> * The feedback in Anyprefer is generated by a reward model, which evaluates preference pairs and guides the optimization of prompts for both the target model and the judge model. Inspired by TextGrad[1], the feedback used to optimize prompts can be interpreted as gradients that update the parameters of the target model (input prompts) and the judge model (prompts to intergrate tool information).
>
> In the revised version of the manuscript, we have clarified this process.
> ****
> > **Q3**: The authors should include their benchmarking on IMG/MED datasets with the synthetic dataset VLFeedback in the main text. If I understand correctly, it similarly does not require human effort, is larger than Anyprefer-V1, and could be compared against to demonstrate the claim made by Figure 6 that the diversity of Anyprefer is its strength over existing techniques. Additionally, I was not convinced by the t-SNE plots that Anyprefer seemed to cover all datasets, it is just in the distribution diagram spread that they appear either more sparse (Anyprefer) or more dense (VLFeedback). A quantitative analysis here consistent with what is done in the literature seems necessary.
>
> **A3**: Thank you for your question. VLFeedback is indeed a relatively new and large-scale preference dataset. In the original manuscript, we have already included a comparison with it in the Natural Vision-Language Understanding domain (Table 6). We also provide the results on Medical Image Analysis in Table R5. As shown, compared to VLFeedback, Anyprefer still demonstrates advantages across various benchmarks. This provides evidence that the quality of data constructed by Anyprefer is superior to that of VLFeedback.
>
> Regarding the t-SNE plots, we follow [2] to use t-SNE technique to evaluate the diversity of the generated preference data and other baselines. A broader and more dispersed distribution, represents greater diversity in the dataset.  To further assess the diversity of the synthesized data, we examine the condition number of the sample matrix. A smaller condition number indicates more diverse data. The results are presented in Table R6, where the smaller condition number of Anyprefer provides additional support for the claim that the Anyprefer dataset exhibits a more diverse coverage. To further validate the diversity of the data, we employed a condition number-based approach to evaluate the diversity of the synthesized data (VLFeedback and Anyprefer-v1). Specifically, we randomly sampled 500 examples from each dataset to construct a covariance matrix of the data matrix (i.e., a matrix of sample x feature) and calculated its condition number to quantify data diversity. A smaller condition number indicates that the data distribution in the embedding space is more dispersed, reflecting greater diversity. As shown in Table R6, the condition number of the Anyprefer dataset is smaller, further supporting the conclusion that the Anyprefer dataset exhibits higher diversity coverage.

---

> > ### Comment · Reviewer_xwba · 2024-11-25
> > **Response to (3/8)**
> >
> > Thank you for the clarification on Q2 with A2. I think the additional prose you provide here helps at least me understand better the optimization process.
> >
> > Regarding the additional analysis of the distribution of generated data, I think that these new results are more convincing as an argument for VLFeedback's diversity.

---

> ### Author Response · Authors · 2024-11-22
> **Response to Reviewer xwba (4/8)**
>
> **Table R5**: Comparison with VLfeedback on medical image analysis.
> | **Method**                | **VQA-RAD (Closed)** | **VQA-RAD (Open)** | **SLAKE (Closed)** | **SLAKE (Open)** | **BLEU-1** | **BLEU-2** | **BLEU-3** | **BLEU-4** | **ROUGE-L** | **METEOR** |
> |---------------------------|---------------------|--------------------|--------------------|------------------|------------|------------|------------|------------|-------------|------------|
> | LLaVA-Med                | 63.57              | 32.09             | 61.30             | 44.26           | 10.31      | 0.66       | 0.07       | 0.01       | 10.32       | 10.95      |
> | + VLFeedback |  64.33   |  32.38   |     61.52    |  44.03  |   10.65   |   0.67    |     0.10  |  0.03 | 10.78  | 11.36
> | + Anyprefer (Ours)              | **72.06**          | **36.10**         | **70.39**         | **49.04**       | **16.85**  | **5.57**   | **2.07**   | **0.56**   | **23.69**   | **29.66**  |
>
> **Table R6**: Condition number comparison on the covariance matric
> | **Method**       | **Condition Number** |
> |-------------------|----------------------|
> | VLFeedback        | 1560.70             |
> | Anyprefer-v1      | **1390.15**         |
> ****
> > **Q4**: The authors claim in Lines 70 and 95 that the methods used as 'tools' for providing auxiliary information to the judge models are 'selected based on the input prompt' or 'Anyprefer strategically selects tools'. From what I understand in section 3.1, either the tools are selected manually corresponding to the task or it is not explained how 'Anyprefer' selects tools. This makes me believe that how the tools are chosen is ambiguous.
>
> **A4**: Thank you very much for raising this question. We apologize for not explaining this clearly in the earlier version. In the updated manuscript, we have clarified this in Section 3.1 and Appendix C. Specifically, we predefine the tools listed for each domain in Section 3.1 (with the flexibility to create new classes for additional tools if needed). For tool selection, given the input data, we use an agent to determine which tools are most suitable for the corresponding input data. In this work, the agent is GPT-4o, which selects tools using a predefined prompt provided in Appendix C. The tools are maintained in a dictionary format as {tool1: info; tool2: info…}, where each tool name corresponds to a brief description. Once the tools are selected, the agent returns the names of the tools as strings, which are then used to extracting information.
>
> ****
> > **Q5**: It would've been interesting to see ablations regardless the dataset generation process. For example, if the Judge model produces quality rankings, then doesn't it seem suboptimal to use only the best and worst generation? Couldn't one use many different pairs (preserving ranking order for + or -) from the ranking? Also it seems that 5 was chosen as the number of generations, but it's unclear the role that this plays in the final quality.
>
> **A5**: Thank you for the reviewer’s thoughtful question. To address your concerns, we conducted additional ablation experiments as follows:
>
> For using more pairs from the rankings, we explored not only using the best and worst generations but also selecting multiple pairs from the Judge model’s rankings (e.g., top-rank 1 vs. middle-rank 3, middle-rank 3 vs. bottom-rank 5) while preserving the ranking order as positive and negative preference pairs. Here, we denote "top-rank 1 vs. middle-rank 3" and "middle-rank 3 vs. bottom-rank 5" as "middle + top" and "middle + bottom". Due to time constraints, we conducted only one iteration of experiments for each of the four domains mentioned in the paper. The experimental results for each domain are presented in Table R7, R8, R9, R10 for natural language generation, nature vision-language understanding, medical image analysis and visuo-motor control, respectively. From the experimental results across several domains, we found that using the rank-1 response as preference data yields better results compared to using the rank 3 response. This highlights the importance of ensuring high-quality preference data. Additionally, we observed that using rank 3 data as dispreference can also be problematic. Even though rank 3 responses are worse than rank 1, they are still better than rank 5 responses. Treating them as dispreference may lead the model to optimize away some originally correct responses, resulting in performance degradation.

---

> ### Author Response · Authors · 2024-11-22
> **Response to Reviewer xwba (5/8)**
>
> **Table R7**: Additional ablation study about preference pair selection on natural language generation:
> | **Method**                   | **GSM8K** | **ARC-Easy** | **ARC-Challenge** | **Alpaca Eval 2.0**  |
> |------------------------------|-----------|--------------|-------------------|----------------------|
> | Llama-2                      | 22.44     | 74.33        | 57.68            | 5.20 / 4.57          |
> | + Self Rewarding             | 23.20     | 74.45        | 56.31            | 3.28 / 3.12          |
> | + Self Rewarding*            | 27.22     | 73.53        | 56.66            | -                    |
> | + Anyprefer (middle + bottom)| 33.18     | 77.12        | 61.42            | 16.12 / 13.48        |
> | + Anyprefer (middle + top)   | 36.42     | 80.03        | 63.15            | 18.47 / 14.75        |
> | + Anyprefer                  | **38.14** | **80.26**    | **64.68**        | **19.25 / 15.14**    |
>
> **Table R8**: Additional ablation study about preference pair selection on natural vision-language understanding:
>
> | **Method**                  | **MME^P** | **MME^C** | **LLAVA^W** | **MMB** | **MMVet** | **SQA^I** | **VisWiz** | **GQA** | **POPE** |
> |-----------------------------|-----------|-----------|-------------|---------|-----------|-----------|------------|---------|----------|
> | LLaVA-1.5-7B               | **1510.7**| 348.2     | 63.4        | 64.3    | 30.5      | 66.8      | 50.0       | 62.0    | 85.90    |
> | + Vfeedback                | 1432.7    | 321.8     | 62.1        | 64.0    | 31.2      | 66.2      | 52.6       | **63.2**| 83.72    |
> | + Human-Prefer             | 1490.6    | 335.0     | 63.7        | 63.4    | 31.1      | 65.8      | 51.7       | 61.3    | 81.50    |
> | + POVID                    | 1452.8    | 325.3     | 68.7        | 64.9    | 31.8      | 68.8      | 53.6       | 61.7    | 86.90    |
> | + RLHF-V                   | 1489.2    | 349.4     | 65.4        | 63.6    | 30.9      | 67.1      | **54.2**   | 62.1    | 86.20    |
> | + Self Rewarding           | 1505.6    | 362.5     | 61.2        | 64.5    | 31.4      | 69.6      | 53.9       | 61.7    | 86.88    |
> | + Anyprefer (middle + bottom)| 1501.8   | 359.4     | 66.8        | 64.8    | 32.2      | 69.1      | 53.7       | 62.0    | 86.75    |
> | + Anyprefer (middle + top) | 1508.3    | 361.2     | 68.5        | 65.0    | 32.8      | 70.1      | 53.8       | 62.2    | 86.89    |
> | + Anyprefer                | 1510.1    | **362.9** | **69.2**    | **65.1**| **33.0**  | **70.9**  | 54.0       | 62.2    | **86.98**|
>
> **Table R9**: Additional ablation study about preference pair selection on medical image analysis:
>
> | **Method**                | **VQA-RAD (Closed)** | **VQA-RAD (Open)** | **SLAKE (Closed)** | **SLAKE (Open)** | **BLEU-1** | **BLEU-2** | **BLEU-3** | **BLEU-4** | **ROUGE-L** | **METEOR** |
> |---------------------------|---------------------|--------------------|--------------------|------------------|------------|------------|------------|------------|-------------|------------|
> | LLaVA-Med                | 63.57              | 32.09             | 61.30             | 44.26           | 10.31      | 0.66       | 0.07       | 0.01       | 10.32       | 10.95      |
> | + Self Rewarding          | 64.17              | 33.29             | 61.30             | 42.63           | 9.71       | 0.97       | 0.10       | 0.01       | 10.38       | 10.52      |
> | + Self Rewarding*         | 66.25              | 32.19             | 63.28             | 42.80           | 9.56       | 1.03       | 0.18       | 0.02       | 11.14       | 11.83      |
> | + Anyprefer (middle + bottom) | 68.19         | 34.22             | 66.74             | 46.19           | 12.47      | 3.12       | 0.84       | 0.15       | 18.92       | 21.12      |
> | + Anyprefer (middle + top) | 70.15            | 35.41             | 68.32             | 47.83           | 14.58      | 4.37       | 1.56       | 0.32       | 20.53       | 25.47      |
> | + Anyprefer               | **72.06**          | **36.10**         | **70.39**         | **49.04**       | **16.85**  | **5.57**   | **2.07**   | **0.56**   | **23.69**   | **29.66**  |

---

> ### Author Response · Authors · 2024-11-22
> **Response to Reviewer xwba (6/8)**
>
> **Table R10**: Additional ablation study about preference pair selection on visuo-motor control.
>
> | **Method**                 | **Put Spoon on Towel (Grasp Spoon)** | **Success** | **Put Carrot on Plate (Grasp Carrot)** | **Success** | **Stack Cube (Grasp Cube)** | **Success** | **Put Eggplant in Basket (Grasp Eggplant)** | **Success** |
> |----------------------------|--------------------------------------|----------------------------------|---------------------------------------|-----------------------------------|----------------------------|--------------------------|---------------------------------------------|-------------------------------------|
> | RT-1                       | 0.10                                | 0.06                             | 0.20                                  | 0.12                              | 0.22                      | 0.02                    | 0.06                                        | 0.00                                |
> | Octo-small                 | 0.42                                | 0.28                             | 0.30                                  | 0.16                              | 0.42                      | 0.10                    | 0.48                                        | 0.32                                |
> | Octo-base                  | 0.38                                | 0.20                             | 0.22                                  | 0.10                              | 0.24                      | 0.04                    | 0.46                                        | 0.32                                |
> | OpenVLA-SFT (baseline)     | 0.46                                | 0.28                             | 0.38                                  | 0.30                              | 0.38                      | 0.14                    | 0.52                                        | 0.32                                |
> | + Self Rewarding*          | 0.50                                | 0.28                             | 0.38                                  | 0.30                              | 0.38                      | 0.14                    | 0.54                                        | 0.34                                |
> | + Anyprefer (middle + bottom) | 0.52                             | 0.35                             | 0.44                                  | 0.36                              | 0.55                      | 0.20                    | 0.60                                        | 0.41                                |
> | + Anyprefer (middle + top) | 0.54                                | 0.38                             | 0.47                                  | 0.40                              | 0.57                      | 0.24                    | 0.63                                        | 0.46                                |
> | + Anyprefer                | **0.56**                            | **0.40**                         | **0.54**                              | **0.44**                          | **0.60**                  | **0.28**                | **0.68**                                    | **0.50**                            |
> ****
> **Q6**: Once the preference datasets are generated using any of the proposed techniques, what was the process for actually improving the target models? We are given the results but without any experimental setup for that tuning. I'm leaning on making this a weakness, as I think it's important to be included, but it could be that this should be a common knowledge process.
>
> **A6**: Thank you for the reviewer’s question and suggestion. First, let us clarify the process. After generating the preference datasets Anyprefer, we performed preference fine-tuning using the DPO [3] method with the constructed preference data. This process yields a stronger model, which we then replace as the target model in Anyprefer. Then, the enhanced target model collaborates with the judge model to generate new preference data, which is subsequently used to further fine-tune the target model. This iterative process can be repeated across multiple rounds. We have include the preference optimization process in Section 2.5 and included the detailed parameter settings for training with DPO in Appendix A of the revised paper.
> ****

---

> ### Author Response · Authors · 2024-11-22
> **Response to Reviewer xwba (7/8)**
>
> **Q7**: Why is it the case that both the prompt of the target model AND of the judge model need to be optimized in this process? Is it not sufficient to optimize the prompt of the target model through the preferences of the judge? It would be interesting to see that this is insufficient.
>
> **A7**: Thank you for the reviewer’s question. This is a very interesting topic. To better demonstrate the necessity of optimizing both prompts simultaneously, we conducted an ablation study with the following ablation models:
> - (1) Optimizing only the input prompt for the target model using the reward.
> - (2) Optimizing only the prompt for the judge model using the reward.
>
> The results of four domains are reported in Table R11 - R14 respectively. The results suggest that jointly updating the judge model and the reward model leads to greater performance improvements than optimizing them individually, further indicating the effectiveness of Anyprefer.
>
> **Table R11**: Ablation study on natural language generation.
> | **Method**                  | **GSM8K** | **ARC-Easy** | **ARC-Challenge** | **Alpaca Eval 2.0**  |
> |-----------------------------|-----------|--------------|-------------------|----------------------|
> | (1) Optimize Target Model Only | 28.12     | 76.02        | 59.12            | 14.58 / 12.34        |
> | (2) Optimize Judge Model Only  | 29.25     | 77.56        | 60.45            | 15.32 / 13.12        |
> | **Anyprefer (Ours)** | **38.14** | **80.26**    | **64.68**        | **19.25 / 15.14**    |
>
> **Table R12**: Ablation study on natural vision-language understanding.
> | **Method**                  | **MME^P** | **MME^C** | **LLAVA^W** | **MMB** | **MMVet** | **SQA^I** | **VisWiz** | **GQA** | **POPE** |
> |-----------------------------|-----------|-----------|-------------|---------|-----------|-----------|------------|---------|----------|
> | (1) Optimize Target Model Only | 1480.2    | 350.2     | 65.2        | 63.9    | 31.0      | 67.2      | 52.0       | 61.5    | 86.10    |
> | (2) Optimize Judge Model Only  | 1485.6    | 353.1     | 66.1        | 64.1    | 31.5      | 68.0      | 53.2       | 61.8    | 86.45    |
> | **Anyprefer (Ours)** | **1510.1**| **362.9** | **69.2**    | **65.1**| **33.0**  | **70.9**  | **54.0**   | **62.2**| **86.98**|
>
>
> **Table R13**: Ablation study on medical image analysis.
> | **Method**                | **VQA-RAD (Closed)** | **VQA-RAD (Open)** | **SLAKE (Closed)** | **SLAKE (Open)** | **BLEU-1** | **BLEU-2** | **BLEU-3** | **BLEU-4** | **ROUGE-L** | **METEOR** |
> |---------------------------|---------------------|--------------------|--------------------|------------------|------------|------------|------------|------------|-------------|------------|
> | (1) Optimize Target Model Only |  66.20   |   33.53   |     65.36    |  45.79  |   12.89   |   3.16    |     1.20  |  0.28 | 14.87  | 17.99
> | (2) Optimize Judge Model Only  | 68.71 |  34.39   |   67.48   |   46.60  |   13.37    |    3.79  |   1.42   |   0.36  |   17.89  | 19.60
> | **Anyprefer (Ours)**              | **72.06**          | **36.10**         | **70.39**         | **49.04**       | **16.85**  | **5.57**   | **2.07**   | **0.56**   | **23.69**   | **29.66**  |
>
> **Table R14**: Ablation study on visuo-motor control.
> | **Method**                     | **Grasp Spoon** | **Success (Spoon)** | **Grasp Carrot** | **Success (Carrot)** | **Grasp Cube** | **Success (Cube)** | **Grasp Eggplant** | **Success (Eggplant)** |
> |--------------------------------|-----------------|---------------------|------------------|----------------------|----------------|--------------------|--------------------|------------------------|
> | (1) Optimize Target Model Only |      0.53       |        0.34         |      0.46        |         0.28         |      0.57      |        0.22        |        0.60        |          0.42          |
> | (2) Optimize Judge Model Only  |      0.53       |        0.36         |      0.49        |         0.30         |      0.58      |        0.24        |        0.62        |          0.44          |
> | **Anyprefer (Ours)**              | **0.56**       |     **0.40**       |     **0.54**     |       **0.44**       |    **0.61**    |      **0.28**      |      **0.66**      |        **0.50**        |
> ****
>
> > **Q8**: Where does so much improvement come from in the medical VQA task? It seems like 31.05% is a massive gain.
>
> **A8**: We sincerely appreciate the reviewer’s insightful question. The significant improvement is due to the fact that current generalist medical LVLMs, pre-trained on large-scale medical image-text datasets, have only grasped some basic medical facts and lack sufficient image-based diagnostic capabilities. As a result, their performance on specialized medical imaging tasks is suboptimal [4,5,6,7].Therefore, leveraging the preference dataset constructed with Anyprefer for preference fine-tuning can enhance the factual accuracy of the model's medical diagnosis and significantly improve its performance.
>
> ****

---

> ### Author Response · Authors · 2024-11-22
> **Response to Reviewer xwba (8/8)**
>
> > **Q9**: In section 3.5, how were these samples manually selected?
>
> **A9**: We sincerely thank the reviewer for raising this question. We have revised the description in the updated paper to address this issue. To clarify, we randomly sampled 200 examples: 100 samples were generated using tool-captured knowledge and feedback mechanisms, while the other 100 were generated without these mechanisms.
> ****
> > **Q10**: Was there an analysis done on how including different numbers of tools was helpful for different datasets, or which tools mattered the most?
>
> **A10**: This is an excellent suggestion. To address it, we conducted an ablation study in the domain of Natural Vision-Language Understanding and Medical Image Analysis to analyze the contribution of different tools to this domain. Specifically, we have added detailed results in Appendix B and referenced them in the main text under the experimental section. We reported the results in Table R15 and Table R16, respectively. According to the results, for both tasks, tools capable of extracting entity-level information (e.g., disease detection) from images outperform description-based tools. Furthermore, combining multiple tools proves to be more effective than relying on a single tool, as it enables the utilization of richer and more diverse information during the generation of preference data. This underscores the importance of integrating the complementary capabilities of different tools.
>
> **Table R15**: Effect of different tool combination on natural vision-language understanding.
> | **Method**                  | **MME^P** | **MME^C** | **LLAVA^W** | **MMB** | **MMVet** | **SQA^I** | **VisWiz** | **GQA** | **POPE** |
> |-----------------------------|-----------|-----------|-------------|---------|-----------|-----------|------------|---------|----------|
> | LLaVA-1.5-7B               | **1510.7**| 348.2     | 63.4        | 64.3    | 30.5      | 66.8      | 50.0       | 62.0    | 85.90    |
> | + Anyprefer (Florence-2-large)           | 1468.3    | 347.8     | 67.0        | 64.2    | 31.7      | 68.6      | 53.4       | 61.9    | 86.52    |
> | + Anyprefer (BLIP-2)                     | 1465.2    | 343.5     | 66.4        | 64.0    | 31.3      | 68.1      | 53.0       | 61.6    | 86.40    |
> | + Anyprefer (Grounded SAM)               | 1475.3    | 355.6     | 68.9        | 64.7    | 32.8      | 69.8      | **54.8**   | 62.0    | 86.75    |
> | + Anyprefer                | 1510.1    | **362.9** | **69.2**    | **65.1**| **33.0**  | **70.9**  | 54.0       | **62.2**| **86.98**|
>
> **Table R16**: Effect of different tool combination on medical image analysis.
> | **Method**                | **VQA-RAD (Closed)** | **VQA-RAD (Open)** | **SLAKE (Closed)** | **SLAKE (Open)** | **BLEU-1** | **BLEU-2** | **BLEU-3** | **BLEU-4** | **ROUGE-L** | **METEOR** |
> |---------------------------|---------------------|--------------------|--------------------|------------------|------------|------------|------------|------------|-------------|------------|
> | LLaVA-Med                | 63.57              | 32.09             | 61.30             | 44.26           | 10.31      | 0.66       | 0.07       | 0.01       | 10.32       | 10.95      |
> | Anyprefer (CheXagent) |  68.02  |  34.74  | 63.72      | 45.80  | 12.82     |  2.10   |  0.89   | 0.23 | 15.80 | 17.42
> | Anyprefer (MedVInT)  | 67.55 |  33.70 |   63.87  | 46.72   |  14.44   |  3.42    |  1.30   |  0.29  | 18.57   | 19.09
> | Anyprefer (MiniGPT-Med) | 69.06  | 35.48   |   64.50     |  46.61  |  13.10   |   3.99   |  1.46   |  0.35  |  20.30  | 22.17
> | + Anyprefer              | **72.06**          | **36.10**         | **70.39**         | **49.04**       | **16.85**  | **5.57**   | **2.07**   | **0.56**   | **23.69**   | **29.66**  |
>
>
> ****
> **Reference**:
>
> [1] Yuksekgonul, M., Bianchi, F., Boen, J., Liu, S., Huang, Z., Guestrin, C., & Zou, J. “TextGrad: Automatic ‘Differentiation’ via Text.” arXiv preprint arXiv:2406.07496 (2024).
>
> [2] Zhao, W., Ren, X., Hessel, J., Cardie, C., Choi, Y., & Deng, Y. “Wildchat: 1M ChatGPT interaction logs in the wild.” arXiv preprint arXiv:2405.01470 (2024).
>
> [3] Rafailov, R., Sharma, A., Mitchell, E., Manning, C. D., Ermon, S., & Finn, C. “Direct preference optimization: Your language model is secretly a reward model.” Advances in Neural Information Processing Systems, 36 (2024).
>
> [4] Chen J, Yang D, Wu T, et al. Detecting and Evaluating Medical Hallucinations in Large Vision Language Models[J]. arXiv preprint arXiv:2406.10185, 2024.
>
> [5] Xia P, Chen Z, Tian J, et al. CARES: A Comprehensive Benchmark of Trustworthiness in Medical Vision Language Models. NeurIPS, 2024.
>
> [6] Royer C, Menze B, Sekuboyina A. Multimedeval: A benchmark and a toolkit for evaluating medical vision-language models[J]. arXiv preprint arXiv:2402.09262, 2024.
>
> [7] Jiang Y, Chen J, Yang D, et al. MedThink: Inducing Medical Large-scale Visual Language Models to Hallucinate Less by Thinking More[J]. arXiv preprint arXiv:2406.11451, 2024.

---

> > ### Comment · Reviewer_xwba · 2024-11-25
> > **Response to the authors.**
> >
> > I heavily thank the authors for their absolutely thorough and complete response to my questions and concerns. I think that essentially all of my original concerns with the paper has been answered in depth. With the vast set of new results and additional prose explaining their method, I believe this the method would be a valuable contribution to subfield of preference data generation--ultimately providing an interesting distillation technique from incredibly powerful models like GPT4-o. One note I maintain is that t-SNE is a contentious tool for demonstrating data-diversity as how it groups data is not always indicative of the data itself [1]. Regardless, I am raising my score to an 8.
> >
> > [1] Yang et al. 'T-SNE Is Not Optimized to Reveal Clusters in Data' https://arxiv.org/abs/2110.02573

---

> ### Author Response · Authors · 2024-11-26
> **Additional Response to Reviewer xwba's Response to (1/8)**
>
> Dear Review xwba,
>
> Thank you for your response. Our goal in formulating this problem as a two-player cooperative game is to provide a prototype framework for understanding the preference data synthesis process. The formulation is motivated by the fact that the universal goal: maximizing the quality of the preference data, cannot be simply achieved by solely improving the target model (generative data) and judge model (generate preference). This goal requires collaboration between the target model and the judge model in generating the response and scoring them. Together, they produce a higher-quality preference dataset than either model could achieve independently. Furthermore, the iterative feedback loop from the reward model ensures mutual benefit and stability in their collaboration. We also provide empirical evidence supporting this cooperative formulation and highlight the importance of collaboration. We hope to leverage more insights from game theory to enhance our model design for preference data synthesis in the future.

---

> ### Author Response · Authors · 2024-11-26
> **Additional Response to Reviewer xwba**
>
> Dear Reviewer xwba,
>
> We sincerely appreciate your detailed and insightful feedback and are glad to see that our response has addressed your concerns. We agree that using t-SNE to evaluate data diversity is not comprehensive enough. While we proposed additional methods during the rebuttal period, further exploration is indeed necessary in the future. Thank you once again for your valuable comments, which have greatly helped us improve our paper.

---

### Official Review · Reviewer_jmiU · 2024-11-04

**Soundness:** 3
**Presentation:** 3
**Contribution:** 3
**Rating:** 6
**Confidence:** 3

**Summary:**

The paper introduces a framework that automates the synthesis of high-quality preference data. It employs a target model and a judge model, and leverages external tools and a feedback mechanism to enhance data quality. The framework generates a diverse dataset, Anyprefer-V1, which contains 58,000 preference pairs, and improves model performance across applications in natural language generation, vision-language understanding, medical image analysis, and visuo-motor control.

**Strengths:**

1. Extensive experiments on four downstream tasks demonstrate that the framework can generalize across various domains and tasks.
2. Besides proposing a framework, the authors also provide their generated synthetic dataset, which would help future works.
3. Their experimental results are promising in all the four downstream tasks.

**Weaknesses:**

1. The overall idea of tool-enhanced critiquing and self-correction is similar to the paper [1]. The authors should have some discussions here.
2. Equation 2 of prompt optimization is misleading. I believe there should not be any gradient descent for the prompt optimization part.
3. In line 196 in Section2.3, the authors should explain how is "high-quality" defined and what kind of reward R(y+,y-) would indicate high-quality reward.
4. The authors only consider self-rewarding as the baseline. It's better to explore and include more baselines, such as [2].

[1] Gou, Zhibin, et al. "Critic: Large language models can self-correct with tool-interactive critiquing." arXiv preprint arXiv:2305.11738 (2023).

[2] Wu, Tianhao, et al. "Meta-rewarding language models: Self-improving alignment with llm-as-a-meta-judge." arXiv preprint arXiv:2407.19594 (2024).

**Questions:**

1. The authors indicate the the self-rewarding method uses the model itself to reward the responses, which would amplify inherent biases. In this paper, the authors use three different models (target model, judge model, and reward model) for this process. However, there can also be inherent biases in any of those three models, why won't this method amplify the inaccuracies and compromise data quality?

---

> ### Author Response · Authors · 2024-11-22
> **Response to Reviewer jmiU (1/3)**
>
> Thank you for your valuable feedback to help us improve our paper. We have revised our paper based on your feedback. We detail our response below and please kindly let us know if our response addresses your concerns.
> ****
> > **Q1**: The overall idea of tool-enhanced critiquing and self-correction is similar to the paper [1]. The authors should have some discussions here. [1] Gou, Zhibin, et al. "Critic: Large language models can self-correct with tool-interactive critiquing." arXiv preprint arXiv:2305.11738 (2023).
>
> **A1**: Thank you pointing out this related work. Critic [1] proposed a tool-interactive critiquing framework, which leverages external tools to iteratively refine the model outputs. The judge model of Anyprefer shares the idea of using tools to refine the model outputs. However, the difference lies on the purpose of applying tools, our method aims to synthesize higher quality preference data, while Critic mainly focuses on refining outputs for self improvement. Despite this, we want to emphasize that tools enhanced refinement is only a part of our framework. The major contribution of this paper is the proposed pipeline that includes data sampling, tools-enhanced refinement, feedback mechanism and etc. We have added this discussion in the related work to acknowledge the sharing idea of tools-enhanced refinement which is used by the Judge model of Anyprefer.
> ****
> > **Q2**: Equation 2 of prompt optimization is misleading. I believe there should not be any gradient descent for the prompt optimization part.
>
> **A2**: Thank you for your question. We ackowledge that the gradient is not actually passing through the prompt optimization. However, the optimization process of our prompts refinement by feedback is similar to updating the parameters via gradient descent. This concept is drawn from TextGrad[2], and we will revise related descriptions to make it more clear. The detailed optimization process is explained as follows:
>
> * In Anyprefer, we optimize both the input prompt of the target model and the tool information utilized by the judge model through a feedback mechanism. This process indirectly updates the target model's policy to generate candidate responses while enhancing the judge model's ability to integrate tool information effectively.
> * The feedback in Anyprefer is generated by a reward model, which evaluates preference pairs and guides the optimization of prompts for both the target model and the judge model. Inspired by TextGrad [2], the feedback used to optimize prompts can be interpreted as gradients that update the parameters of the target model (input prompts) and the judge model (prompts to intergrate tool information).
>
> In the revised version of the paper, we have clarified this process.
> ****
> > **Q3**: In line 196 in Section2.3, the authors should explain how is "high-quality" defined and what kind of reward $R(y+,y-)$ would indicate high-quality reward.
>
> **A3**: Thank you for the suggestion. Our definition of high-quality data is as follows: for a pair of preference data, the preferred data should align correctly with the user-provided input prompts, while the non-preferred data should be incorrect in the same context. Moreover, the overall difference between the preferred and non-preferred data should not be significant. In other words, the distinction between non-preferred and preferred data should be difficult to discern, as this is crucial for the model to learn effectively during training [3]. Thus, we designed a specific prompt (detailed in Appendix C.2) to guide the reward model in evaluating the quality of the constructed preference data. We input the synthesized preference pair ($y+$, $y-$) into the reward model, which generates a reward score $R(y+,y-)$ on a scale of 1 to 10. A higher score indicates higher-quality preference data.
> ****
>
> > **Q4**: The authors only consider self-rewarding as the baseline. It's better to explore and include more baselines, such as Meta-rewarding[3].
>
> **A4**: Thank you for your suggestion. Due to the inapplicability of the prompt design in Meta-rewarding for the Visuo-Motor Control domain, we have added Meta-rewarding as a baseline in the new version of the manuscript for the other three domains: Natural Language Generation, Natural Vision-Language Understanding, and Medical Image Analysis. Additionally, we have discussed this method in the experiment and related work section. We report the results in the following three Tables (Table R1 - R3). According to the results, we observe that Anyprefer outperforms the meta-rewarding method in all tasks. Though Meta-rewarding improved the rewarding bias issue of self-rewarding, it is inferior than Anyprefer incorporating external tools and learning from feedback mechanism to mitigate this issue.

---

> ### Author Response · Authors · 2024-11-22
> **Response to Reviewer jmiU (2/3)**
>
> **Table R1**: Results on natural language generation.
> | **Method**               | **GSM8K** | **ARC-Easy** | **ARC-Challenge** | **Alpaca Eval 2.0** |
> |--------------------------|-----------|--------------|-------------------|---------------------|
> | LLaMA-2                 | 22.44     | 74.33        | 57.68            | 5.20 / 4.57         |
> | + Self Rewarding        | 23.20     | 74.45        | 56.31            | 3.28 / 3.12                  |
> | + Meta Rewarding | 25.47     | 76.22        | 59.47            | -                   |
> | + Anyprefer              | **38.14** | **80.26**    | **64.68**        | **19.25 / 15.14**   |
>
> **Table R2**: Results on natural vision-language understanding.
>
> | **Method**           | **MME^P** | **MME^C** | **LLAVA^W** | **MMB** | **MMVet** | **SQA^I** | **VisWiz** | **GQA** | **POPE** |
> |-----------------------|-----------|-----------|-------------|---------|-----------|-----------|------------|---------|----------|
> | LLaVA-1.5-7B         | **1510.7**| 348.2     | 63.4        | 64.3    | 30.5      | 66.8      | 50.0       | 62.0    | 85.90    |
> | + Self Rewarding      | 1505.6    | 362.5     | 61.2        | 64.5    | 31.4      | 69.6      | 53.9       | 61.7    | 86.88    |
> | + Meta Rewarding      | 1498.3    | 357.4     | 64.0        | 64.2    | 31.3      | 69.1      | 53.5       | 62.0    | 86.70    |
> | + Anyprefer           | 1510.1    | **362.9** | **69.2**    | **65.1**| **33.0**  | **70.9**  | **54.0**       | **62.2**    | **86.98**|
>
> **Table R3**: Results on medical image analysis.
> | **Method**            | **VQA-RAD (Closed)** | **VQA-RAD (Open)** | **SLAKE (Closed)** | **SLAKE (Open)** | **BLEU-1** | **BLEU-2** | **BLEU-3** | **BLEU-4** | **ROUGE-L** | **METEOR** |
> |------------------------|---------------------|--------------------|--------------------|------------------|------------|------------|------------|------------|-------------|------------|
> | LLaVA-Med             | 63.57              | 32.09             | 61.30             | 44.26           | 10.31      | 0.66       | 0.07       | 0.01       | 10.32       | 10.95      |
> | + Self Rewarding      | 64.17              | 32.29             | 61.30             | 42.63           | 9.71       | 0.97       | 0.10       | 0.01       | 10.38       | 10.52      |
> | + Meta Rewarding       | 67.42              | 33.05             | 65.10             | 45.12           | 12.48      | 1.23       | 0.24       | 0.03       | 12.56       | 13.21      |
> | + Anyprefer            | **72.06**          | **36.10**         | **70.39**         | **49.04**       | **16.85**  | **5.57**   | **2.07**   | **0.56**   | **23.69**   | **29.66**  |
> ****
> > **Q5**: The authors indicate the the self-rewarding method uses the model itself to reward the responses, which would amplify inherent biases. In this paper, the authors use three different models (target model, judge model, and reward model) for this process. However, there can also be inherent biases in any of those three models, why won't this method amplify the inaccuracies and compromise data quality?
>
> **A5**: In self-rewarding, the model assigns scores to its own responses. Throughout the process, there is no constraint on the self-generated judgment, thus leading to the inherent rewarding bias issue[4]. Anyprefer has advantage in mitigating such bias issue:
>
> * First, we propose to use external tools to mitigate such inherent bias issue, since it is believed the multiple tools can provide more accurate information than the model itself. Specifically, the judge model incorporates selected tool information as a reference to assist in ranking the preferences, and the reward model subsequently assigns the final reward score. This approach, which combines multiple sources of information, should be more accurate than relying solely on the model’s own judgment.
>
> * We acknowledge that each involved model can also have its own bias, however, Anyprefer uses the combination of a target model, a judge model, and a reward model is actually kind of ensemble, which can effectively minimize the inherent bias of each single model. This is also confirmed by existing research on ensemble rewarding[5,6,7].
> ****

---

> ### Author Response · Authors · 2024-11-22
> **Response to Reviewer jmiU (3/3)**
>
> **Reference**:
>
> [1] Gou, Zhibin, et al. "Critic: Large language models can self-correct with tool-interactive critiquing." arXiv preprint arXiv:2305.11738 (2023).
>
> [2] Yuksekgonul, M., Bianchi, F., Boen, J., Liu, S., Huang, Z., Guestrin, C., & Zou, J. “TextGrad: Automatic ‘Differentiation’ via Text.” arXiv preprint arXiv:2406.07496 (2024).
>
> [3] Wu, Tianhao, et al. "Meta-rewarding language models: Self-improving alignment with llm-as-a-meta-judge." arXiv preprint arXiv:2407.19594 (2024).
>
> [4] Xu, Wenda, et al. "Pride and prejudice: LLM amplifies self-bias in self-refinement." Proceedings of the 62nd Annual Meeting of the Association for Computational Linguistics (Volume 1: Long Papers). 2024.
>
> [5] Coste, Thomas, et al. "Reward model ensembles help mitigate overoptimization." arXiv preprint arXiv:2310.02743 (2023).
>
> [6] Zhang, Shun, et al. "Improving reinforcement learning from human feedback with efficient reward model ensemble." arXiv preprint arXiv:2401.16635 (2024).
>
> [7] Eisenstein, Jacob, et al. "Helping or herding? reward model ensembles mitigate but do not eliminate reward hacking." arXiv preprint arXiv:2312.09244 (2023).

---

> > ### Author Response · Authors · 2024-11-27
> > **Kindly Request Feedback from Reviewer jmiU**
> >
> > Dear Reviewer jmiU,
> >
> > Thank you once again for your insightful feedback! We’d like to follow up regarding the ICLR discussion phase. Your guidance has been invaluable, and we’ve made every effort to address your concerns and improve the paper accordingly. As the paper revision deadline approaches, we would greatly appreciate knowing if the changes resolve your concerns.
> >
> > We understand you have a busy schedule, but if you could take the time to share your latest comments, we would be deeply grateful. Your feedback is crucial for refining our work further, and we welcome any additional discussion.
> >
> > Thank you for your time and expertise in helping us enhance our research!

---

> ### Comment · Reviewer_jmiU · 2024-11-27
>
> Thank you for the authors' detailed responses. I've raised my score to 6.

---

### Meta-Review · Area_Chair_9s32 · 2024-12-23

**Metareview:**

Summary:
========
The paper presents Anyprefer, a framework for generating synthetic preference data by framing it as a two-player Markov game between a target model and a judge model (GPT-4o). The framework incorporates external tools and a feedback mechanism to optimize prompts. The authors demonstrate its effectiveness through Anyprefer-V1, a dataset of 58,000 preference pairs, showing improvements across natural language generation, vision-language understanding, medical image analysis, and visuo-motor control tasks.

Strengths
=======
* Broad applicability demonstrated across multiple domains and tasks
* Integration of external tools with judge model
* Strong experimental results compared to baselines
* Release of Anyprefer-V1 dataset for community use

Weaknesses (mostly before rebuttal):
==========
* Insufficient theoretical justification for the two-player Markov game framework
* Unclear explanation of tool selection process
* Lack of clarity on prompt optimization methodology
* Missing quantitative analysis of dataset diversity vs. alternatives like VLFeedback
* No clear definition of "high-quality" preferences

**Additional Comments On Reviewer Discussion:**

There were good engagements between the reviewers and the authors to address the concerns about:
* Framing the method as two-play game which is not really a technical aspect,
* How tools were selected for tasks
* More baselines
* Clarity of prompt optimization details
* Connection between tool use and prompt optimization

The authors responded by proving new experimental results, clarifying concepts. The responses were relatively well-received, although some concerns remain (e.g., game framing).

---

### Decision · Program_Chairs · 2025-01-22

Accept (Poster)